# LIMANS: Linear Model of the Adversarial Noise Space

## Abstract

Recent works have revealed the vulnerability of deep neural network (DNN) classifiers to adversarial attacks. Among such attacks, it is common to distinguish specific attacks adapted to each example from universal ones referred to as example-agnostic. Even though specific adversarial attacks are efficient on their target DNN classifier, they struggle to transfer to others. Conversely, universal adversarial attacks suffer from lower attack success. To reconcile universality and efficiency, we propose a model of the adversarial noise space that allows us to frame specific adversarial perturbation as a linear combination of the universal adversarial directions. We bring in two stochastic gradient-based algorithms for learning these universal directions and the associated adversarial attacks. Empirical analyses conducted on the CIFAR-10 and ImageNet datasets show that LIMANS (i) enables crafting specific and robust adversarial attacks with high probability, (ii) provides a deeper understanding of DNN flaws, and (iii) shows significant ability in transferability.

## 1 Introduction

With recent technological advances, deep neural networks (DNN) are widespread in numerous applications ranging from biomedical imaging to autonomous vehicles. However, DNNs are vulnerable to adversarial attacks (Szegedy et al., 2014). The latter are slight perturbations of *clean* examples well classified by the DNN, leading to misclassification. These perturbations may take the form of common corruptions (e.g., for images, it can be a change in lightning conditions, colorimetry, or rotations) or visually imperceptible learned adversarial noises.

There essentially exist two ways of crafting adversarial noises. The first strategy consists in finding a paired adversarial noise with each example to attack (Croce et al., 2021; Qian et al., 2022). As such, it is deemed to be *specific* since each adversarial noise is *specifically* designed for a given example. The second strategy aims at finding a unique *universal* noise which, added to any example, is likely to fool the DNN (Moosavi-Dezfooli et al., 2017). Each strategy comes with its pros and cons. On the one hand, although specific attacks achieve great performances on the target DNN, the learned adversarial noises do not fool other DNNs on the same examples. They transfer poorly. On the other hand, universal attacks have shown great transferability at the expense of a weaker ability to fool the target DNN on which the universal adversarial noise is learned.

To reconcile specificity and universality, we propose a way to model the space of adversarial noises. This space is supposed to be embedded in the span of the ensemble of directions, perpendicular to the decision boundaries (Li et al., 2020). Since the dimensionality of such spanned space depends on the classifier's decision boundaries, it is likely to lie in a low dimensional manifold and so does the adversarial noise space. This leads to considering a linear model of the adversarial noise space. In addition, it has been shown in (Tramèr et al., 2017) that the decision boundaries of multiple DNN classifiers trained on the same dataset are close to each other. This leads us to think that a model of the adversarial noise space could be transferable.

The present work proposes to bridge the gap between specific and universal attacks by linear modeling the Adversarial Noise Space (LIMANS). Intuitively, the dimension of this model should range between 1, for universal attacks, and the dimension of examples, in the case of specific attacks. The overall idea is sketched in Figure 1. For each example to attack, an adversarial noise is crafted as a linear combination of adversarial directions. While the adversarial directions are *universal*, the linear combination coefficients are *specific* to each example to perturb.

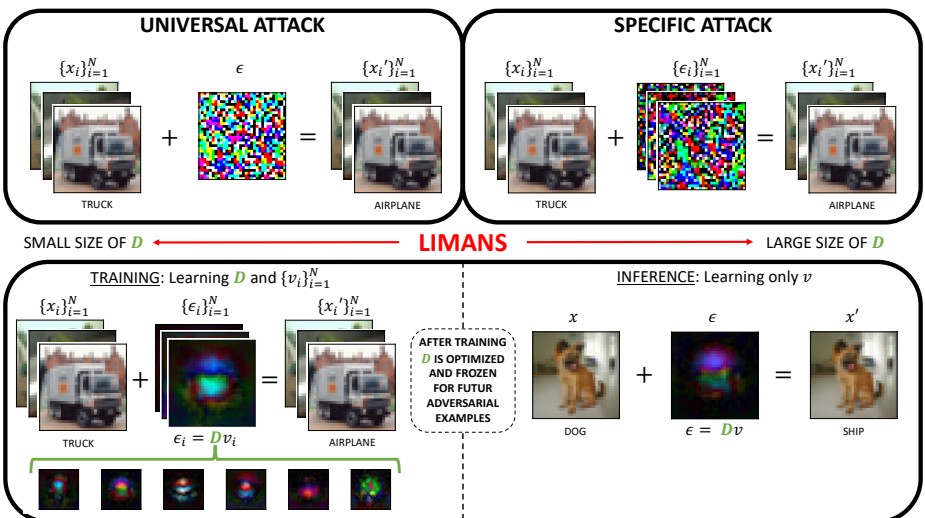

Figure 1: High level overview of the proposed LIMANS adversarial attack and its optimization. It is highlighted the adversarial model $D$ is universal to every adversarial example, while a specific coding vector $v$ is tailored to each adversarial example.

In short, the main contributions of the paper are:

- LIMANS, a model of adversarial perturbation as a linear combination of universal adversarial directions that are visually inspectable, helps to better understand the DNN flaws.

- The associated optimization problem, allows us to learn these universal adversarial directions and two relaxations leading to scalable stochastic gradient-based algorithms.

- Empirical evidence illustrating that adversarial examples generated by LIMANS are more robust to existing adversarial example detectors than current state-of-the-art adversarial attacks.

- Experiments demonstrating that the learned ensemble of adversarial directions is transferable across different classifiers achieving state-of-the-art transferability.

The rest of the paper is organized as follows: the state-of-the-art research on specific attacks, universal attacks, and manifold of adversarial perturbations are presented in Section 2. The optimization problem of LIMANS and the proposed algorithmic solutions are detailed in the Section 3. Finally, Section 4 displays the adversarial noise model and provides experimental evaluations of the adversarial noise space in terms of robustness and transferability on both CIFAR10 and ImageNet.

## 2 BRIDGING THE GAP BETWEEN SPECIFIC AND UNIVERSAL ADVERSARIAL ATTACKS

Consider a classifier $f$, typically a DNN to be attacked. Threat models can be expressed as a function $g$ such that $g(f, \mathbf{x}) = \mathbf{x}'$, called adversarial example, aims to fool $f$ for a given input $\mathbf{x}$. Adversarial examples are typically crafted by adding an adversarial perturbation to the input under attack. It is common to distinguish *example-based perturbations* also called *specific perturbations* from *universal perturbations* which are *example-agnostic*.

**Specific attacks.** Specific attacks are designed to generate, given a classifier $f$ and for each input example, an associated adversarial perturbation (see (Croce et al., 2021; Qian et al., 2022) and references therein for a detailed review). The most popular ones include one-shot gradient method such as the fast gradient sign method (FGSM) and more elaborated iterative procedures such as the projected gradient descent (PGD), DeepFool or the method by Carlini and Wagner (CW). Last but not least, AutoAttack (Croce & Hein, 2020), an ensemble of diverse parameter-free attacks, is becoming the state-of-the-art for evaluating the robustness of a neural network. The specificity of these attacks enables them to fool the targeted classifier with high probability. However, it has been

found that they are not as effective in fooling other classifiers, namely, adversarial examples yielded by specific attacks are poorly transferable from one classifier to another. To enhance the transferability of specific attacks, (Xie et al., 2019) proposed adopting inputs transformation, while (Wang & He, 2021) suggested stabilizing the update directions of adversarial noise to escape poor local optima, leading to the methods VNI-FGSM and VMI-FGSM. Besides, the Neuron Attribution-based Attacks, NAA, boosted the transferability by conducting the feature-level attacks (Zhang et al., 2022). Finally, the attack generating reverse adversarial perturbation, RAP, attempted to find adversarial attacks located on a plateau of the loss function (Qin et al., 2022).

**Universal attacks.** A specific attack requires, for each new example, to solve an optimization problem that fully hinges on $f$. To overcome this issue, Universal Adversarial Perturbation (UAP), an example-agnostic perturbation, was introduced in (Moosavi-Dezfooli et al., 2017). It consists of a single perturbation which, added to any examples, leads to fooling $f$ with high probability. Later on, different extensions of UAP were proposed: UAP-PGD (Shafahi et al., 2020) used a gradient-based algorithm termed to compute UAP, CD-UAP (Zhang et al., 2020) optimized a universal perturbation on a given subset of classes while the Class-Wise Universal Adversarial Perturbations (CW-UAP) (Benz et al., 2021) elaborated a universal perturbation per class. Universal attacks are fast but their applicability is still far-fetched because of poor performances compared to specific attacks (Chaubey et al., 2020). These adversarial perturbations are carefully crafted estimating the worst-case performance of a classifier. Besides, there is another type of universal perturbation the so-called common corruptions (i.e. Gaussian Noise, Impulse Noise...), which exists in real-world imaging systems (Hendrycks & Dietterich, 2019). They are average-case perturbations beyond the scope of this work. Even though common corruptions are closer to real-world harm, they need to be explicitly defined before any perturbation crafting which limits to modelling of the perturbation space. In trying to bridge both the norm-bounded adversarial perturbations world and the common corruptions world, researchers tried to learn the manifold in which adversarial perturbations are embedded.

**Learning the manifold of adversarial perturbations.** Works in this direction aim at giving the reason why adversarial attacks exist and design defense methods to overcome them. Some focus on researching the space of the adversarial noise to an input example. It has been demonstrated firstly the space was a large continuous region (Tabacof & Valle, 2016). Then, (Tramèr et al., 2017) discovered that the decision boundaries of different classifiers are close and proposed to establish a transferable subspace of the space across different classifiers. However, the space is inferred based on the adversarial noise generated by the FGSM method which impacts the precision of the found space. The hypothesis of the transferability depending only on the dimensionality of this space limited its performance on CNN classifiers. On the other hand, some studies on the overall structure of the classifier clarify the rise of adversarial attacks. Research in (Fawzi et al., 2018) illustrated the decision boundaries of a classifier are in the vicinity of examples and flat in most the directions. More recently, (Li et al., 2020) claimed that adversarial noise is caused by gradient leakage and the adversarial directions are perpendicular to the classifier boundaries. Based on the above works, in this paper, we propose to learn universal adversarial directions, adapted to the dataset while independent from a single input example under attack, spanning the adversarial noise space. With this space, it is thus allowed to retrieve specific adversarial examples.

## 3 LINEAR MODELING OF THE ADVERSARIAL NOISE SPACE (LIMANS)

In this section, we start by introducing our framework for modeling the adversarial noise space and, we propose two algorithmic schemes to address the problem.

### 3.1 PROBLEM SETTING

Let $f : \mathbb{R}^P \rightarrow \mathbb{R}^c$ be a DNN classifier which outputs $f(\mathbf{x}) \in \mathbb{R}^c$, the vector of scores for an example $\mathbf{x} \in \mathcal{X} \subset \mathbb{R}^P$ to belong to a class of $y \in \mathcal{Y} = \{1, \cdots, c\}$. The predicted class is given by $\operatorname{argmax}_k f_k(\mathbf{x})$. Given an example $\mathbf{x}$, an adversarial attack model seeks an adversarial perturbation $\boldsymbol{\epsilon}$ such that $\mathbf{x}' = \mathbf{x} + \boldsymbol{\epsilon}$, the adversarial example, is a valid example *i.e.* $\mathbf{x}' \in \mathcal{X}$, close to $\mathbf{x}$ and induces $\operatorname{argmax}_k f_k(\mathbf{x}) \neq \operatorname{argmax}_k f_k(\mathbf{x}')$. As the perturbation must be indiscernible, customary one enforces $\|\boldsymbol{\epsilon}\|_p \leq \delta_p$ for some $\ell_p$-norm (typically $\ell_2$ and $\ell_\infty$) and some small $\delta_p \in \mathbb{R}_+$. Specific adversarial attacks learn perturbations $\boldsymbol{\epsilon} = \boldsymbol{\epsilon}(\mathbf{x})$, *i.e.* dedicated to a specific example $\mathbf{x}$ while at the opposite universal attacks such as UAP seek a single perturbation $\boldsymbol{\epsilon}$ able to attack any test examples.

Specifically, the originality of the work is to express the adversarial noise paired to $\mathbf{x}$ as $\boldsymbol{\epsilon}(\mathbf{x}) = D\mathbf{v}(\mathbf{x})$ where $D \in \mathbb{R}^{P \times M}$ is a dictionary composed of M normalized adversarial noise atoms and $\mathbf{v}(\mathbf{x}) \in \mathbb{R}^M$ a coding vector (further on we simply write $\mathbf{v}$ for readability). While the dictionary $D$ is shared across the examples, the coding vector $\mathbf{v}$ is specifically crafted for any given $\mathbf{x}$. By setting atomNumber $= 1$, the learned adversarial perturbation becomes universal, while setting atomNumber $= P$ (the dimension of the examples) results in a specific adversarial perturbation.

Given a trained DNN classifier $f$, LIMANS consists of a training stage where the dictionary $D$ is learned using a labeled set $\mathcal{T} = \{(\mathbf{x}^{(i)}, y^{(i)})\}_{i=1}^{N}$ and an inference stage where, given $D$, and any new example $\mathbf{x}^{(k)}$, the corresponding coding vector $\mathbf{v}^{(k)}$ is crafted to make $D\mathbf{v}^{(k)}$ an adversarial perturbation of $\mathbf{x}^{(k)}$. Notice that as M $\ll$ P, the searching space of the LIMANS attacks is a low dimensional space (spanned by the atoms) which is much lower than the original space $\mathcal{X}$. We frame the learning procedure of LIMANS as maximizing the fooling rate under the constraints listed above.

**Problem 3.1** (LIMANS formulation). Given the classifier $f$ and the training set $\mathcal{T} = \{(\mathbf{x}^{(i)}, y^{(i)})\}_{i=1}^{N}$, find $D \in \mathbb{R}^{P \times M}$ and $V \in \mathbb{R}^{M \times N}$ solution of

$$
\max_{\substack{D \in \mathbb{R}^{P \times M} \\ V \in \mathbb{R}^{M \times N}}} \quad \sum_{i=1}^{N} \mathbb{1}\{\operatorname*{argmax}_{k} f_k(\mathbf{x}^{(i)'}) \neq \operatorname*{argmax}_{k} f_k(\mathbf{x}^{(i)})\},
$$

$$
s.t. \quad \begin{cases} \mathbf{x}^{(i)'} = \mathbf{x}^{(i)} + D\mathbf{v}^{(i)} \in \mathcal{X} & i = 1, \dots, N, \\ ||D\mathbf{v}^{(i)}||_p \leq \delta_p & i = 1, \dots, N, \\ ||D_j||_p = 1 & j = 1, \dots, M, \end{cases} \tag{1}
$$

where $\mathbb{1}_A$ denotes the indicator function of the set $A$.

### 3.2 Algorithmic schemes

The indicator function $\mathbb{1}$ being non-smooth and highly non-convex, instead of maximizing the fooling rate it is usual to consider minimizing a surrogate loss function $L_\gamma(f(\mathbf{x}'), f(\mathbf{x}))$, parameterized by $\gamma$, more amenable to optimization. Given an original example $\mathbf{x}$ classified as $\operatorname{argmax}_k f_k(\mathbf{x}) = y$, typical loss function of interest is

$$
\mathcal{L}_\gamma(f(\mathbf{x}'), f(\mathbf{x})) = \max\big(-\gamma, \; f_y(\mathbf{x}') - \max_{k \neq y} f_k(\mathbf{x}')\big), \tag{2}
$$

Still, the problem is hard to solve because of the non-convexity of the constraint $||D\mathbf{v}^{(i)}||_p \leq \delta_p$ or the account for the constraint $\mathbf{x}^{(i)'} \in \mathcal{X}$. One way to tackle this issue is to rely on the nonconvex proximal splitting framework (Sra, 2012; Rakotomamonjy, 2013) that happened to be less effective than expected in practice. Instead, a stochastic gradient approach turned out to be fruitful. To implement it, the optimization problem 1 has to be relaxed to handle the constraints in a differentiable way in the objective function of Eq. 1. Hence, for computational tractability, we hereafter propose two relaxations of 3.1, Simple-LIMANS and Regularized-LIMANS along with their respective solver.

**Regularized-LIMANS** The first relaxation is a regularized version expressed as follows.

$$
\min_{\substack{D \in \mathbb{R}^{P \times M} \\ V \in \mathbb{R}^{M \times N}}} \sum_{i=1}^{N} \mathcal{L}_\gamma(f(\mathbf{x}^{(i)} + D\mathbf{v}^{(i)}), f(\mathbf{x}^{(i)})) + \lambda h_{(\delta_p, p)}(D, \mathbf{v}^{(i)}) \quad s.t. \; D \in \mathcal{D} \tag{3}
$$

with $\lambda \in \mathbf{R}_+$ a regularisation parameter, $\mathcal{D} = \{D \mid ||D_j||_p = 1, \forall j \in \{1, \dots, M\}\}$ and $h_{(\delta_p, p)}$ representing a penalty function. We consider the $\ell_p$-norm, with $p = 2$ or $p = \infty$, as penalty function leading to $h_{(\delta_2, 2)}(D, \mathbf{v}) = \max(||D\mathbf{v}||_2 - \delta_2, 0)$ and $h_{(\delta_\infty, \infty)}(D, \mathbf{v}) = \sum_k \max(|(D\mathbf{v})_k| - \delta_\infty, 0)$.

Here, we get rid of the constraints $\mathbf{x}^{(i)'} \in \mathcal{X}, \forall i$ and enforce small magnitude of $||D\mathbf{v}^{(i)}||_p$ through the regularizer $h_{(\delta_p, p)}$. Empirically this promotes $\mathbf{x}^{(i)} + D\mathbf{v}^{(i)} \in \mathcal{X}$ to be nearly close to $\mathcal{X}$. Algorithm 1 summarizes the optimization scheme of Regularized-LIMANS.

The Regularized-LIMANS optimizes $(D, V)$ in a stochastic fashion, and specifically, $D$ is updated using a projected gradient descent that ensures that the constraints $||D_j||_p = 1, \forall j$ are satisfied.

---

**Algorithm 1** Regularized-LIMANS

**Require:** Classifier $f$; Learning rate $\rho$; Training dataset $\mathcal{T}$; $\ell_p$ budget $\delta_p$; Optimizer Optim; Batch size B; Regularization parameter $\lambda$

1: $D = \mathcal{N}(0, \mathbb{1}_{\mathrm{M} \times \mathrm{P}})$; $V = \mathcal{N}(0, \mathbb{1}_{\mathrm{P} \times \mathrm{M}})$
2: **for** k = 0 to MAXEPOCH **do**
3:     loss = 0
4:     **for** $(\mathbf{x}^{(i)}, y^{(i)}) \subset \mathcal{T}$ **do**
5:        $\mathbf{x}^{(i)'} = \mathbf{x}^{(i)} + D\mathbf{v}^{(i)}$
6:        $\hat{y}_{adv} = f(\mathbf{x}^{(i)'})$;    $\hat{y} = f(\mathbf{x}^{(i)})$
7:        $\mathrm{loss}_i = \mathcal{L}_0(\hat{y}_{adv}, \hat{y}) + \lambda h_{(\delta_p, p)}(D, \mathbf{v}^{(i)})$
8:        loss = loss + loss$_i$
9:        **if** modulo$(i) = B$ **then**
10:          $D \leftarrow \mathrm{Optim}(\nabla_D \mathrm{loss})$      (Update)
11:          $V \leftarrow \mathrm{Optim}(\nabla_V \mathrm{loss})$      (Update)
12:          $D = \mathrm{Proj}_{\{D \,|\, \|D\|_p = 1\}}(D)$
13:          loss = 0
14:        **end if**
15:     **end for**
16: **end for**
17: $D\mathbf{v}^{(i)} \leftarrow \mathrm{Proj}_{\{D\mathbf{v} \,|\, \|D\mathbf{v}\|_p \leq \delta\}}(D\mathbf{v}^{(i)})$
18: $\mathbf{x}^{(i)'} \leftarrow \mathrm{Proj}_{\mathcal{X}}(\mathbf{x}^{(i)} + D\mathbf{v}^{(i)})$
19: **return** $\{\mathbf{x}^{(i)'}\}_{i=1}^{\mathrm{N}}, (D, V)$

---

**Algorithm 2** Simple-LIMANS

**Require:** Classifier $f$; Learning rate $\rho$; Training dataset $\mathcal{T}$; $\ell_p$ budget $\delta_p$; Optimizer Optim; Batch size $B$

1: $D = \mathcal{N}(0, \mathbb{1}_{\mathrm{M} \times \mathrm{P}})$; $V = \mathcal{N}(0, \mathbb{1}_{\mathrm{P} \times \mathrm{M}})$
2: **for** k = 0 to MAXEPOCH **do**
3:     loss = 0
4:     **for** $(\mathbf{x}^{(i)}, y^{(i)}) \subset \mathcal{T}$ **do**
5:        $\mathrm{noise}^{(i)} = D\mathbf{v}^{(i)}$
6:        $\mathbf{x}^{(i)'} = \mathrm{proj}_{\mathcal{X}}(\mathbf{x}^{(i)} + \frac{\delta_p \mathrm{noise}^{(i)}}{\|\mathrm{noise}^{(i)}\|_p})$
7:        $\hat{y}_{adv} = f(\mathbf{x}^{(i)'})$;    $\hat{y} = f(\mathbf{x}^{(i)})$
8:        $\mathrm{loss}_i = \mathcal{L}_\infty(\hat{y}_{adv}, \hat{y})$
9:        loss = loss + loss$_i$
10:        **if** modulo$(i) = B$ **then**
11:          $D \leftarrow \mathrm{Optim}(\nabla_D \mathrm{loss})$      (Update)
12:          $V \leftarrow \mathrm{Optim}(\nabla_V \mathrm{loss})$      (Update)
13:          loss = 0
14:        **end if**
15:     **end for**
16: **end for**
17: $V \leftarrow [\|D_{\bullet j}\|_p V_{j\bullet}] \,\forall j \in \{1, \ldots, M\}$
18: $D \leftarrow \mathrm{Proj}_{\mathcal{D}}(D)$
19: **return** $\{\mathbf{x}^{(i)'}\}_{i=1}^{\mathrm{N}}, (D, V)$

---

A grid search on $\lambda$ allows to control the generalization of the model. In practice, for the selected value of $\lambda$, it happens that the constraint on $\|D\mathbf{v}\|_p$ is slightly violated. To ensure the respect of the constraint, if needed a post-processing is performed. The stochastic optimization makes Regularized-LIMANS applicable to large-scale datasets such as ImageNet. Details on the tuning of the hyper-parameters involved in this optimization scheme are provided in the supplementary material.

**Simple-LIMANS** The Regularized-LIMANS requires the tuning of the hyper-parameter $\lambda$ which may be cumbersome. To alleviate that, we propose a second relaxation of LIMANS that involves an objective function encompassing two of the constraints, the last one being taken care of by post-processing. Specifically, the method termed Simple-LIMANS solves the following problem:

$$\min_{\substack{D \in \mathbb{R}^{\mathrm{P} \times \mathrm{M}} \\ V \in \mathbb{R}^{\mathrm{M} \times \mathrm{N}}}} \sum_{i=1}^{\mathrm{N}} \mathcal{L}_\gamma(f(\mathrm{proj}_{\mathcal{X}}(\mathbf{x}^{(i)} + \frac{\delta_p D\mathbf{v}^{(i)}}{\|D\mathbf{v}^{(i)}\|_p})), f(\mathbf{x}^{(i)})) \tag{4}$$

where $\mathrm{proj}_{\mathcal{X}}$ denotes the projection operator that maps its input $\mathbf{x}^{(i)'} = \mathbf{x}^{(i)} + D\mathbf{v}^{(i)}$ onto $\mathcal{X}$. Simple-LIMANS trades off the constraint $D \in \mathcal{D}$, *i.e.* the unit norm constraint over the atoms of $D$, for the explicit guarantee that the adversarial example $\mathbf{x}^{(i)'}$ is valid (i.e. belongs to $\mathcal{X}$) and that the adversarial noise is utmost of magnitude $\delta_p$ by defining $\mathbf{x}_i^{(i)'}$ as: $\mathbf{x}^{(i)'} = \mathrm{proj}_{\mathcal{X}}(\mathbf{x}^{(i)} + \frac{\delta_p D\mathbf{v}^{(i)}}{\|D\mathbf{v}^{(i)}\|_p})$. Here $\mathrm{proj}_{\mathcal{X}}$ is the projection operator onto $\mathcal{X}$. Simple-LIMANS solves (4) by iteratively updating $D$ and $V$ using a gradient descent procedure as shown in Algorithm 2. It proves computationally efficient as it does not require hyper-parameter tuning. At termination, a post-processing is used to ensure $D \in \mathcal{D}$ without changing the adversarial examples.

**Attack of unseen examples** At inference time, given $D$, and an unseen example $\mathbf{x}^{(k)}$, we seek an adversarial counterpart $\mathbf{x}'^{(k)} = \mathbf{x} + D\mathbf{v}^{(k)}$ where $\mathbf{v}^{(k)}$ is computed either with 1 or Algorithm 2 restricted to the optimization of $\mathbf{v}^{(k)}$.

## 4 EXPERIMENTS

This section presents the experimental evaluations of the adversarial noise space and adversarial perturbations generated with it, providing a comparison with the state-of-the-art attacks on benchmark datasets. They consist of two parts. Firstly, we empirically demonstrate the existence of the adversarial noise space and robustness of the generated adversarial noises by adopting the Simple-LIMANS 2. Secondly, we estimate the transferability of the adversarial noise space across different classifiers with the more precise algorithm Regularized-LIMANS 1.

### 4.1 EXPERIMENTAL SETTINGS

Our experiments are conducted on two datasets: CIFAR-10 (Krizhevsky et al., 2009) and ImageNet ILSVRC2012 (Krizhevsky et al., 2017) As suggested in (Zhang et al., 2021), we perform the experiments only on the validation set and split it into three parts, the first set for training the model $D$, the second for the tuning of $\lambda$ when using Regularized-LIMANS and the last one for testing.

**CIFAR-10 Experiments.** The number of examples for training, validation and test is respectively 8000, 1000, and 1000. The experiments on validation of the proposed model and the robustness estimation are conducted on the pre-trained VGG11 with batch normalization and the robust ResNet-18 (Sehwag et al., 2022) classifier. The transferability of the proposed model has been evaluated over 4 vanilla DNNs, i.e., MobileNet-V2, ResNet50, DenseNet121, and the VGG11 as aforementioned, and 2 robust DNNs, robust ResNet-18[1] and robust WideResNet-34-10 [1] (Sehwag et al., 2022). These experiments have been implemented in Pytorch on a MacBook Pro with 2,3 GHz Intel Core i9, 8 cores and a GPU Nvidia RTX 2080.

**ImageNet Experiments.** The number of examples for training, validation and test is respectively 10000, 2000, 5000. We select here the 4 vanilla classifiers, ResNet-18, MobileNet-V2, DenseNet121 and VGG11 and two robust classifiers available on RobustBench[1], robust ResNet-18 and robust WideResNet-50-2 (Salman et al., 2020). The experiments on large scale dataset was performed on a server equipped with 4 GPU Volta V100-SXM2-32GB.

**State-of-art methods.** For a fair comparison on the performance of attacks and their robustness, we consider the $\ell_\infty$ specific attack baselines, AutoAttack, PGD FGSM, and the universal attack baselines UAP-PGD, Fast-UAP and CW-UAP. For comparison on the transferability, it involves the state-of-the-art attacks, VNI-FGSM, NAA and RAP, and the classical one, AutoAttack. The above specific attacks are implemented by resorting to TorchAttacks library (Kim, 2020) which contains Pytorch implementation of the most popular specific attacks. While, the universal attacks are implemented based on the publicly available resources.

**Metric.** The Attack Success Rate, also known as Fooling Rate (FR) that is $\frac{1}{N} \sum_{i=1}^{N} \arg\max_k f_k(\mathbf{x}^{(i)'}) \neq \arg\max_k f_k(\mathbf{x}^{(i)})$, is used to assess the performance of our adversarial attacks. However, the robustness of the attack can be evaluated with the Robust Accuracy Under Defense (RAUD) metric, originally proposed and termed as Attack Success Rate under Defense (ASRD) by (Lorenz et al., 2022), as an unbiased metric of adversarial attacks and measuring the percentage of the successful attacks to a classifier under an adversarial example detector. More details about this metric and the considered detector choices are given in the supplementary material.

Details of the parameter settings and the hyper-parameter selection are given in supplementary material C. We provide also the results of our proposed $\ell_2$-attacks in supplementary material D.

### 4.2 EXPERIMENTAL RESULTS

**Learned adversarial noise space.** Figure 2 tracks the fooling rate of LIMANS for different number of the adversarial atoms under the $\ell_2$ norm constraint (figures of the $\ell_\infty$-attack are given in supplementary material). It shows that the LIMANS attack is always stronger than universal baselines as even for only one atom, that attack allows to tune its coefficient making it more efficient. We see that from $M = 500$ the LIMANS attacks closes the gap with state-of-the-art specific adversarial attacks. This results is interesting as it empirically shows that by tuning the number of atoms $M$, the proposed LIMANS does bridge the gap between specific and universal adversarial attacks. Besides at inference,

---

[1] https://robustbench.github.io/

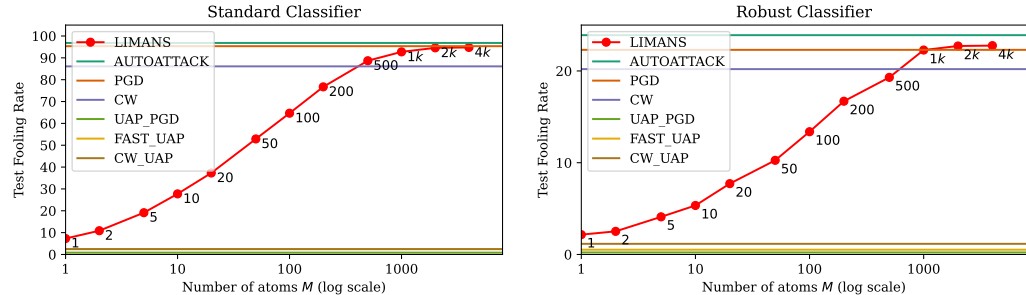

Figure 2: Test fooling rate of adversarial attacks under the $\ell_2$ norm constraint ($\delta_2 = 0.5$) on CIFAR-10 test data when fixing a number of atoms M (x axis), associated to the classifier (left) VGG11 and (right) robust ResNet-18.

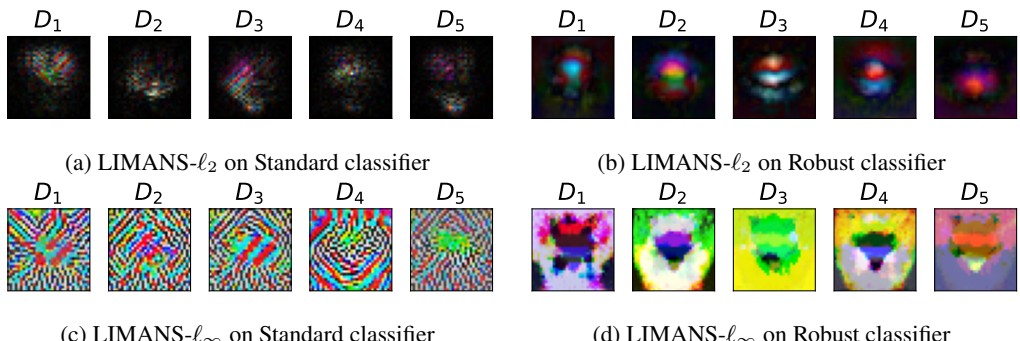

(a) LIMANS-$\ell_2$ on Standard classifier  (b) LIMANS-$\ell_2$ on Robust classifier

(c) LIMANS-$\ell_\infty$ on Standard classifier  (d) LIMANS-$\ell_\infty$ on Robust classifier

Figure 3: Visualization of the learned universal adversarial directions (atoms of the dictionary $D$) when M = 5, on CIFAR-10 and corresponding to the classifier (left) VGG11 and (right) robust ResNet-18. All atoms have been rescaled for display.

by setting M = 500 the LIMANS attacks only optimize for a coding vector $\mathbf{v}$ of dimension 500 whereas specific adversarial attacks here need to optimize 3072 variables. Furthermore such result confirms the manifold hypothesis of the adversarial noise space and confirms the efficiency of the quick, parameter-free algorithm Simple-LIMANS. By setting the model to be linear, LIMANS allows to visually inspect the adversarial information. Figure 3 displays the optimized model's atoms when M = 5. It shows how the atoms constituting the universal base fooling the classifiers are structured. This structure differs according to the classifier and the considered $\ell_p$ norm. In particular, the dictionary of LIMANS-$\ell_2$ on the robust classifier is reminiscent of certain Fourier decompositions.

**Robustness of the attack.** This experiment was conducted with the same settings as in (Lorenz et al., 2022). Table 1 shows the performances of robustness of the proposed $\ell_\infty$-attack and its comparison with the specific adversarial attack baselines. It is noted that the state-of-the-art specific attacks become completely or nearly harmless when classifier is protected by an attack detector. The proposed LIMANS attack surpasses these specific attacks when M is only 10. With M $\geq$ 500, the proposed attack can successfully jeopardize the classical classification system, even equipped with an attack detector. The robust classifier shows its stability facing adversarial attacks while the proposed attack can also ignore the attack detectors and damage the system to some extent. It indicates thus the potential application of the proposed attacks in evaluating the effectiveness of an adversarial example detector plugged prior in a classical or robust classifier.

**Transferable adversarial noise space.** The results in this part were generated using the algorithm Regularized-LIMANS. By adjusting the hyper-parameter $\lambda$, we managed to find an optimized M = 150 for CIFAR-10, which achieves almost comparable performance with AutoAttack and to analyze the transferability of the learned space without loss of precision as shown in Table 2. When it comes to ImageNet (Table 3), considering the large size of an image and the memory limitation, this dimension becomes M = 100. This is far from the real dimensionality of the adversarial noise

Table 1: Robustness performance of the LIMANS $\ell_\infty$-attack ($\delta_\infty = 8/255$) in term of RAUD on the CIFAR-10 test data and against the attack detectors plugged in both standard classifier (S.C.) and robust classifier (R.C.). **The smaller the RAUD, the more robust the adversarial attack is**. The best performance is marked in black bold font.

| Detectors d | $d_{FGSM}$ | | $d_{PGD}$ | | $d_{Autoattack}$ | | $d_{LIMANS_{10}}$ | |
|---|---|---|---|---|---|---|---|---|
| Classifiers $f$ | S.C. | R.C | S.C. | R.C | S.C. | R.C | S.C. | R.C |
| SA | 91.1 | 85.1 | 91.1 | 85.1 | 91.1 | 85.1 | 91.1 | 85.1 |
| FGSM | *91.1* | *85.1* | *91.1* | *85.1* | *91.1* | *85.1* | 83.4 | 79.5 |
| PGD | 90.6 | 84.9 | *91.1* | 85.0 | *91.1* | *85.1* | 55.9 | 73.7 |
| Autoattack | 89.9 | 84.6 | 90.9 | 85.0 | *91.1* | 85.0 | 52.7 | 71.5 |
| LIMANS$_{10}$ | 75.7 | 81.0 | 81.0 | 80.8 | 81.6 | 81.0 | 88.9 | 79.6 |
| LIMANS$_{500}$ | 17.5 | 71.5 | 25.6 | 72.2 | 31.8 | 74.2 | 26.6 | 69.4 |
| LIMANS$_{1000}$ | 15.9 | 70.1 | 26.1 | 70.9 | 32.1 | 72.5 | **21.7** | 68.7 |
| LIMANS$_{4000}$ | **15.6** | **69.6** | **23.7** | **70.4** | **28.2** | **72.6** | 31.1 | **68.4** |

Table 2: Performance of $\ell_\infty$-attacks on CIFAR-10 ($\delta_\infty = 8/255$), in terms of fooling rate (FR) and standard accuracy (SA), where the left column lists the source classifiers and the first line presents the target classifiers. The best results of transferability are marked in red bold style. That of the specific attacks are shown in black bold style.

| | | MobileNet | ResNet50 | DenseNet | VGG | R-r18 | R-wrn-34-10 |
|---|---|---|---|---|---|---|---|
| | AutoAttack | 63.3 | **100** | 54.6 | 25.1 | 1.2 | 2.4 |
| | VNI-FGSM | 78.3 | 95.9 | 80.3 | 57.2 | 2.7 | 2.1 |
| ResNet50 | NAA | 50.7 | 64.7 | 22.9 | 18.4 | 1.4 | 2.1 |
| | RAP | 49.0 | 75.1 | 52.5 | 35.4 | 1.6 | 2.8 |
| | Ours | **96.0** | 91.3 | **81.8** | **82.1** | **11.7** | **13.2** |
| | AutoAttack | 62.5 | 43.0 | 44.0 | **100** | 2.7 | 2.7 |
| | VNI-FGSM | 69.3 | 62.6 | 61.4 | 96.5 | 3.0 | 2.6 |
| VGG | NAA | 42.3 | 14.5 | 1.8 | 71.6 | 1.6 | 1.2 |
| | RAP | 46.5 | 39.5 | 40.9 | 73.8. | 3.3 | 3.4 |
| | Ours | **97.4** | **87.5** | **81.5** | 91.0 | **11.5** | **12.6** |

space, and hence, does not lead to comparable performance with specific attacks such as AutoAttack. However, it still offers evidence for the transferability of the learned space on ImageNet. Moreover, for robust classifiers, the decision boundaries are more complicated, which leads to failure in closing the performance gap between LIMANS attack and AutoAttack when only 100 adversarial directions are learned. Nevertheless, it always shows good performance in transferring.

It is claimed in (Tramèr et al., 2017) that the distance between the decision boundaries of two classifiers trained on the same dataset is small. This means that the adversarial space learned by LIMANS, if it corresponds to the space spanned by the set of directions perpendicular to the decision boundaries, is transferable between different classifiers. The results reported in the Table 2 confirm this intuition. The dictionaries built upon a ResNet50 and a VGG show better transfer performance over state-of-the-art attacks across vanilla classifiers.

However, as shown in Table 3, two exceptions on ImageNet challenge this conclusion. To address them, we propose considering the relative performances. The learned space on a classifier with restricted dimensionality allows us to successfully find adversarial perturbations only for a part of the examples and can be used to attack another classifier achieving comparable performances. Thus, regarding the transferability of LIMANS, we draw the same conclusion on ImageNet across classifiers. Besides, we note that the transferable property also holds between vanilla and robust classifiers. The LIMANS model learned on a vanilla classifier used to fool a robust classifier (and vice versa), gives slightly worse results than the one learned on a classifier of the same category. This might be due to the differences in the dataset used to train the classifiers which result in a larger bias between the decision

Table 3: Performance of $\ell_\infty$-attacks on ImageNet ($\delta_\infty = 4/255$), in terms of FR, where the left column lists the source classifiers and the first line presents the target classifiers. The best results of transferability are marked in red bold font. Those of the specific attacks are shown in black bold font.

| | | MobileNet | ResNet18 | DenseNet | VGG | R-r18 | R-50-2 |
|---|---|---|---|---|---|---|---|
| ResNet18 | AutoAttack | 40.30 | **100** | 35.76 | 34.90 | 1.80 | 1.34 |
| | VNI-FGSM | 56.74 | 99.98 | 51.40 | **51.42** | 2.84 | 2.04 |
| | NAA | 22.54 | 97.94 | 14.84 | 19.30 | 2.12 | 1.20 |
| | RAP | 53.36 | 96.74 | 51.30 | 50.60 | 3.80 | 3.14 |
| | Ours | **59.16** | 59.16 | **53.14** | 48.28 | **10.48** | **6.62** |
| VGG | AutoAttack | 47.94 | 40.06 | 32.62 | **100** | 2.34 | 1.42 |
| | VNI-FGSM | **57.98** | 53.96 | 42.88 | 99.84 | 2.76 | 2.24 |
| | NAA | 19.62 | 14.92 | 12.18 | 79.96 | 2.18 | 1.40 |
| | RAP | 53.14 | 53.12 | 42.68 | 95.68 | 3.48 | 2.84 |
| | Ours | 57.68 | **54.14** | **50.04** | 51.62 | **10.68** | **6.24** |
| R-r18 | AutoAttack | 13.70 | 15.8 | 10.82 | 14.60 | **71.74** | 10.78 |
| | VNI-FGSM | 16.14 | 17.66 | 12.48 | 16.08 | 63.22 | 11.74 |
| | NAA | 11.46 | 10.86 | 9.34 | 11.42 | 21.48 | 4.90 |
| | RAP | 11.32 | 10.80 | 8.16 | 10.32 | 45.80 | 7.94 |
| | Ours | **37.14** | **33.2** | **33.76** | **29.90** | 29.84 | **12.94** |
| R-50-2 | AutoAttack | 20.14 | 22.76 | 17.36 | 19.44 | 15.42 | **59.02** |
| | VNI-FGSM | 23.88 | 26.22 | 19.68 | 23.28 | 18.00 | 52.28 |
| | NAA | 14.08 | 13.12 | 10.20 | 14.04 | 9.82 | 12.58 |
| | RAP | 13.82 | 14.06 | 10.52 | 13.50 | 15.54 | 34.10 |
| | Ours | **42.18** | **42.50** | **42.46** | **34.22** | **23.70** | 18.02 |

boundaries of the two types of classifiers. Yet the performance is still remarkable, e.g., on ImageNet, $FR_{(\text{R-50-2}\rightarrow\text{ResNet-18})} = 78\%$ $FR_{(\text{VGG}\rightarrow\text{ResNet-18})}$ and $FR_{(\text{ResNet-18}\rightarrow\text{R-r18})} = 35\%$ $FR_{(\text{R-r18}\rightarrow\text{R-r18})}$.

Furthermore, it is worth noting that the performance of LIMANS attack on a target classifier does not depend on its performance on the source classifier, but on the nature of this target classifier. In Table 2, the fooling rate of the LIMANS-specific attack on ResNet is $91.3\%$. However, when the learned LIMANS model is used to generate adversarial perturbation to fool MobileNet, its performance is even better reaching $96.0\%$. This is because MobileNet is simpler and easier to attack. Finally, through comparison and analysis, we conclude that a model trained on a robust classifier is more easily transferable to other classifiers.

## 5 CONCLUSIONS

This work introduced LIMANS, a linear model of the adversarial noise space, allowing it to bridge the gap between universal and specific adversarial attacks. It also proposes two implementations, Simple-LIMANS a parameter-free algorithm, and Regularized-LIMANS, more efficient when its regularization parameter is well tuned. For the use of LIMANS, our results suggest starting with Simple-LIMANS to quickly obtain a suitable solution and, depending on the available computation time, improving it by finding a relevant regularization parameter allowing to use of the more accurate Regularized-LIMANS solver. Empirical evidence revealed that adversarial examples crafted by LIMANS were more robust against adversarial examples detectors and proved the adversarial noise space to be transferable leading to results better than than current state-of-the-art adversarial attacks. Up to now, only adversarial noise space trained on a specific DNN in white-box settings has been considered. The next step is to consider LIMANS for generating black-box attacks and training them on multiple DNNs, making them even more universal, efficient, and close to the true adversarial harm in real-life applications.

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
