# Supplementary Material to
# LIMANS: Linear Model of the Adversarial Noise Space

## A  Robust accuracy under defense

In this section we detail the definition of the RAUD and the involved tools we used to compute it.

### A.1  Definition

From an attacker's point of view, the goal is to create the adversarial attack that is the most harmful, i.e., that fools the targeted classifier the most. Therefore, an adversarial attack maximizing the fooling rate is sought. Conversely, from a defender's perspective, the aim is to keep a high classification accuracy even in presence of adversarial examples. Thus, the defender only considers adversarial examples that are not classified as the true label, $\mathrm{argmax}_k f_k(\mathbf{x}') \neq y$, (see Yang et al. (2020) and Wang et al. (2017) for more details).

From this, the robustness can be measured as the number of clean examples that are well classified and for which no adversarial examples can be found in a $\delta$-ball around them. This quantity is estimated by the Robust Accuracy RA as the empirical counterpart of the astuteness Wang et al. (2017) and is defined as

$$\mathrm{RA}(f, \mathcal{D}) = \frac{1}{\mathrm{N}} \sum_{i=1}^{\mathrm{N}} \mathbb{1}\{\mathcal{A}_{\{(\mathbf{x}^{(i)}, y^{(i)})\}} = \varnothing\}, \tag{1}$$

with $\mathcal{A}_{\{(\mathbf{x}^{(i)}, y^{(i)})\}} = \{(\mathbf{x}^{(i)'}, y^{(i)})\}$ being the set of adversarial examples associated to $(\mathbf{x}^{(i)}, y^{(i)})$.

Yet, this later metric is questioned by Lorenz et al. Lorenz et al. (2022) who emphasized on the importance of taking into account the ability to easily detect adversarial examples. To this end, to properly measure the dangerousness of an attack, it has been suggested to train an adversarial example detector $d$ to be used upstream of the classifier $f$. By doing so, they showed that Autoattack adversarial examples can be easily detected, making the attack inefficient. From this observation, they suggest to use another metric, the Robust Accuracy Under Defense (RAUD) which stands as a natural robustness metric from both the attacker and defender point of view,

$$\mathrm{RAUD}(f, \mathcal{D}) = \frac{1}{\mathrm{N}} \sum_{i=1}^{\mathrm{N}} \mathbb{1}\{\forall \mathbf{x}' \in \mathcal{A}_{\{(\mathbf{x}^{(i)}, y^{(i)})\}}, \ d(\mathbf{x}') = 0\}, \tag{2}$$

where $d : \mathbb{R}^{\mathrm{P}} \mapsto \{0, 1\}$ is an adversarial example detector with 0 as the clean example's label.

### A.2  Details about the detector $d$

Regarding the design of a descent adversarial examples detector, we followed the guidelines proposed by Lorenz et al. (2021) and Harder et al. (2021). In these two works, authors propose to use a random forest binary classifier with at least 100 trees, in the Fourier domain either of the input images or of the Fourier Features of Feature-Maps from the images. Authors showed that by using either one of the inputs, the random forest binary classifier is able to discriminate with high precision adversarial example computed from SOTA specific adversarial attacks on several complex datasets such as CIFAR10, CIFAR100, ImageNet or Celeba. In order to lower as much as possible the bias introduced by the detector, we chose to use a random forest binary classifier with 300 trees in the Fourier domain of the input images. Our choice has been confirmed by extensive experiments and is highlighted in Table 4.

Table 4: Confusion matrices of the detectors used to computed the RAUD of Table 1. All detectors have been trained on the same training dataset as the one used in the training of LIMANS and the displayed values of computed over the validation dataset, such that fair performances are considered. TN: True Negative, FP: False Positive, FN: False Negative, TP: True Positive.

| Detectors d | | | $d_{FGSM}$ | | $d_{PGD}$ | | $d_{Autoattack}$ | | $d_{LIMANS_{10}}$ | |
|---|---|---|---|---|---|---|---|---|---|---|
| Confusion Matrix | TN | FP | 863 | 57 | 814 | 106 | 790 | 130 | 846 | 74 |
| | FN | TP | 0 | 920 | 0 | 920 | 0 | 920 | 95 | 825 |
| Accuracy | | | 96.9 % | | 94.2 % | | 92.9 % | | 90.8 % | |
| Precision | | | 94.1 % | | 89.6 % | | 87.6 % | | 91.7 % | |

Indeed Table 4 displays the confusion matrix of the detectors involved in the results presented in Table 1 of the main paper.

The detectors have been trained on the same training dataset as the one used in the training of LIMANS and the displayed values of computed over the validation dataset. Finally the RAUD metric of the different adversarial attacks using the detector is performed on the test dataset, which none has previously seen before making everything totally fair.

We empirically observe highly effective detectors discriminating almost perfectly real images than adversarial images, without mistaking one or the other by producing low False-Positive and False-Negative values. These performances gave use confidence in the use of these detectors as a tool of the RAUD metric, evaluating both the harmfulness and the transferability of adversarial attacks, which are essential to be measure when assessing the quality of an adversarial attack.

## B  VISUALLY INTERPRETABLE ADVERSARIAL PERTURBATIONS

By modelling the adversarial noise space, we empirically observe that LIMANS' parameters capture the most meaningful information fooling the classifier. In some way it can be seen as capturing the semantics of the objects within the classification space. Figure **??** highlights very interesting atoms that indeed spotlight recurring patterns in classification such as edges and corners for the $\ell_\infty$ atoms and local spots in the images for $\ell_2$ atoms. Along with Figure **??** we also show that the same conclusions can be reached for simpler dataset such as MNIST with Figure 4,

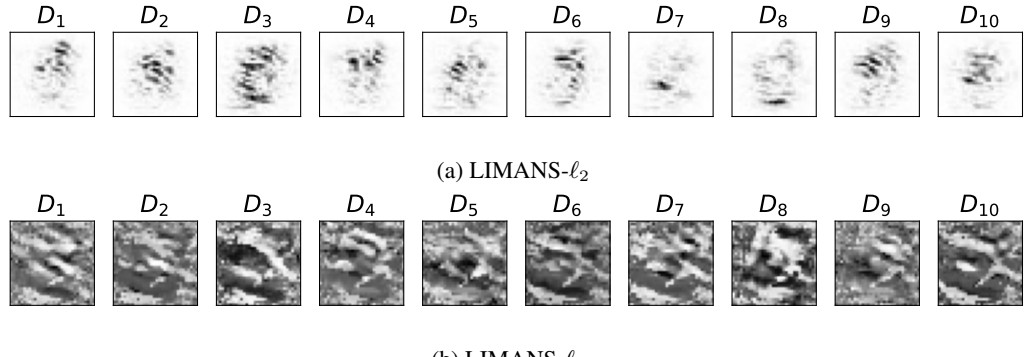

$D_1$  $D_2$  $D_3$  $D_4$  $D_5$  $D_6$  $D_7$  $D_8$  $D_9$  $D_{10}$

(a) LIMANS-$\ell_2$

$D_1$  $D_2$  $D_3$  $D_4$  $D_5$  $D_6$  $D_7$  $D_8$  $D_9$  $D_{10}$

(b) LIMANS-$\ell_\infty$

Figure 4: Visualization of the learned universal adversarial directions (atoms of the dictionary $D$) when M $= 10$, on MNIST according to both the $\ell_2$ and $\ell_\infty$ norm targetting a LeNet classifier achieving more than 98.8% of test accuracy. All atoms have been rescaled for display.

In addition to these visually interesting universal parameters, this empirical observation is furthermore proven when inspecting the LIMANS produced specific adversarial perturbations.

Figure 5 displays LIMANS specific adversarial perturbations on the MNIST dataset (purposely used to ease the observations). It is clear that LIMANS produces much more interesting adversarial perturbations than state-of-the-art specific adversarial perturbations that are absolutely random.

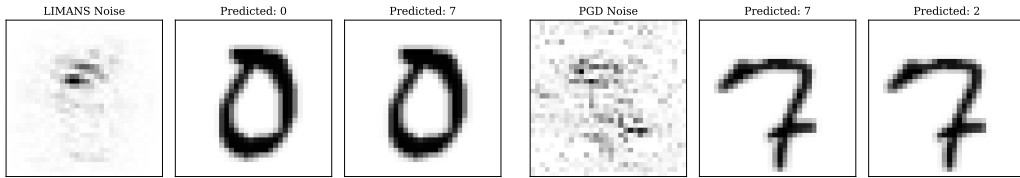

(a) LIMANS adversarial perturbation and example   (b) PGD adversarial perturbation and example

Figure 5: Examples of $\ell_2$ adversarial perturbations produced by LIMANS and PGD on the MNIST dataset for a LeNet classifier achieving more than 98.8% of test accuracy.

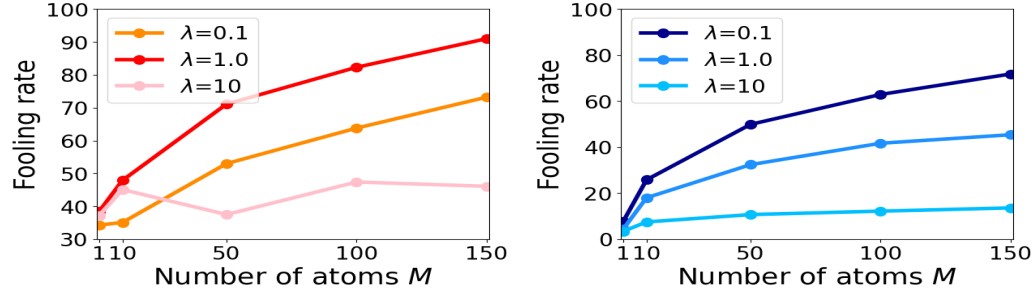

Figure 6: Performance of LIMANS (left) $\ell_\infty$-attacks and (right) $\ell_2$-attacks on CIFAR-10 when attacking VGG, under different settings of hyper-parameter in Regularized LIMANS $\lambda$, and different number of atoms M.

LIMANS however target the most sensitive spots to fool the classifiers, which makes LIMANS adversarial attack a much more realistic attack than state-of-the-art specific adversarial attacks.

## C  EXPERIMENTAL DETAILS

This section presents the details of the different implementations used in both the Regularized-LIMANS and Simple-LIMANS experiments.

### C.1  REGULARIZED-LIMANS HYPER-PARAMETERS AND SETTINGS

**Regularized-LIMANS hyper-parameters**  Figure 6 illustrates the impact of the hyper-parameter $\lambda$ and the number of atoms M on attacking performance. With appropriately increasing M, the performance will be improved, which confirms the conclusion in the paper. In the experiments, considering the trade-off between attacking performance and memory limit, we choose just M = 150 for CIFAR-10 and M = 100 for ImageNet, and do not go further. Additionally, when $p = \infty$, letting $\lambda = 1$ provides the best performance, while $\lambda = 0.1$ is the suitable setting when $p = 2$. This conclusion is valid when extending to other classifiers, as shown in Table 5.

**Experimental settings in transferability estimation**  In order to quantify the transferability of the adversarial noise space, we carried out experiments on CIFAR-10 with 5 vanilla DNNs and 2 robust DNNs, namely, MobileNet, Inception, ResNet, DenseNet, VGG, Robust ResNet18, and Robust WideResNet-34-10, and their respective accuracy are 94.00%, 94.10%, 93.2%, 92.8%, 92.1%, 82.3%, and 85.1%. When on ImageNet, the Inception is not considered due to its different input size, and all classifiers are off-the-shelf in pytorch model zoo and offer respectively the accuracy, 70.95%(MobileNet), 68.20%(ResNet), 73.65%(DenseNet), 67.60%(VGG), 51.25%(Robust ResNet18) and 66.55%(Robust WideResNet-50-2). In the experiments, we used each DNN as the source classifier to learn the adversarial noise space, then, crafted the adversarial perturbation in the learned space to deceive the other classifiers that were regarded as the target classifiers.

Table 5: Performance of LIMANS attacks on CIFAR-10, in terms of FR, when the number of atoms M = 150. The best results are marked in red bold style.

| | $\ell_\infty$-attack | | | $\ell_2$-attack | | |
|---|---|---|---|---|---|---|
| Classifiers $\lambda$ | VGG | MobileNet | R-R18 | VGG | MobileNet | R-R18 |
| 0.1 | 73.2 | 85.3 | 15.8 | **71.7** | **95.4** | **17.6** |
| 1.0 | **91.0** | **97.3** | **25.3** | 45.4 | 49.6 | 12.8 |
| 10 | 46.1 | 91.7 | 19.9 | 13.6 | 24.4 | 10.3 |

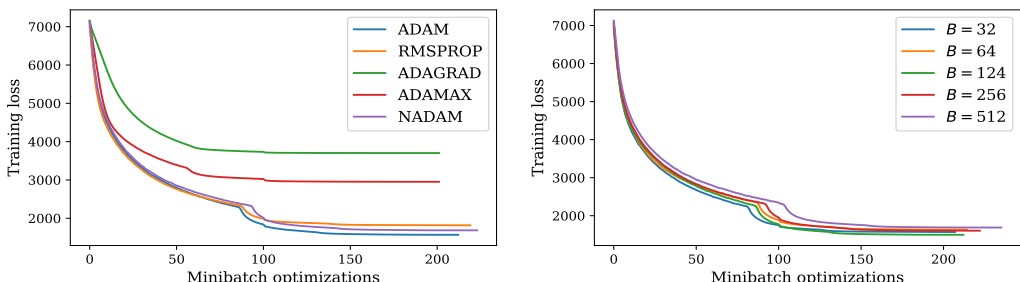

Figure 7: Evolution of LIMANS$_{10}$ training loss according to (left) different optimizers and (right) different batch sizes $B$ using the Simple-LIMANS algorithm on the standard classifier under the $\ell_\infty$ norm.

The other parameters used in the experiments are: In the training phase, the number of iterations is MAXEPOCH=1000. The learning rate $\rho$ is fixed to 0.001. The value of $\delta$ is that used in RobustBench**??**. In the validation and the test phase, MAXEPOCH is 150 when $p = 2$ and 300 when $p = \infty$.

**The state-of-the-art attacks for transferability comparisons** We presented in our paper the comparison with the specific $\ell_\infty$-attack AutoAttack and three state-of-the-art transferable $\ell_\infty$-attacks VNI-FGSM, NAA, and RAP. Besides, we consider also Translation-Invariant Attack (TI-FGSM), VMI-FGSM, and universal $\ell_\infty$-attacks UAP and UAPPGD (The well-known DI-FGSM, MI-FGSM, or NI-FGSM was not selected to compare, as all these processing are already integrated into the attack such as RAP). The LIMANS $\ell_2$-attack was compared with the $\ell_2$ version of AutoAttack, UAP, UAPPGD, and RAP, and the classical $\ell_2$-attack, CW method.

## C.2 SIMPLE-LIMANS HYPER-PARAMETERS

The main idea behind Simple-LIMANS is to optimize the LIMANS parameters the simplest way possible, with the least hyper-parameter tuning and optimization details possible.

**Modeling details** The Simple-LIMANS algorithms operates a relaxation on the definition of the adversarial noise. Given the original example $\mathbf{x}^{(i)}$, Simple-LIMANS consider its adversarial noise as $\boldsymbol{\epsilon}^{(i)} = D\mathbf{v}^{(i)} + \mathbf{b}$ the product of the universal adversarial noise model $D$ with its corresponding coding vector $\mathbf{v}^{(i)}$, to which is added the offset $b$ universal to all LIMANS adversarial noises.

**Learning rate.** No regularization parameter tuning is needed, furthermore, the learning rate is automatically taken care of, by using the scheduler ReduceLROnPlateau, which aims at reducing the learning rate when the loss plateaus. The optimization simply needs to start with a pretty high learning rate which will avoid bad random initialisation local minima, and then will decrease as the optimization proceeds, allowing the parameters to reach a better minimum. The parameters of the used ReduceLROnPlateau are: patience=40, factor=0.1 and threshold=1e-1.

Table 6: Robustness performance of the LIMANS $\ell_\infty$-attack ($\delta_\infty = 8/255$) in terms of RAUD on the CIFAR-10 test data and against the attack detectors plugged in both standard classifier (S.C.) and robust classifier (R.C.). **The smaller the RAUD, the more robust the adversarial attack is**. The best performances are marked in bold red.

| Detectors d | $d_{PGD}$ | | $d_{Autoattack}$ | | $d_{LIMANS_{10}}$ | |
|---|---|---|---|---|---|---|
| Classifiers $f$ | S.C. | R.C | S.C. | R.C | S.C. | R.C |
| SA | 91.1 | 85.1 | 91.1 | 85.1 | 91.1 | 85.1 |
| FGSM | $91.1 \pm 0.0$ | $85.1 \pm 0.0$ | $91.1 \pm 0$ | $85.1 \pm 0.0$ | $89.3 \pm 0.0$ | $79.5 \pm 0.0$ |
| PGD | $91.0 \pm 0.0$ | $84.9 \pm 0.1$ | $91.0 \pm 0.0$ | $85.0 \pm 0.0$ | $80.7 \pm 0.5$ | $73.3 \pm 0.3$ |
| Autoattack | $91.0 \pm 0.0$ | $85.0 \pm 0.0$ | $91.1 \pm 0.0$ | $85.0 \pm 0.0$ | $78.2 \pm 0.3$ | $71.5 \pm 0.3$ |
| LIMANS$_{10}$ | $78.3 \pm 2.4$ | $79.6 \pm 0.0$ | $81.7 \pm 2.0$ | $8.0 \pm 0.0$ | $86.4 \pm 1.6$ | $79.5 \pm 0.1$ |
| LIMANS$_{500}$ | $26.3 \pm 0.3$ | $71.3 \pm 0.1$ | $32.1 \pm 0.4$ | $73.2 \pm 0.2$ | $36.5 \pm 3.1$ | $69.4 \pm 0.2$ |
| LIMANS$_{1000}$ | $24.7 \pm 0.9$ | $70.4 \pm 0.2$ | $31.6 \pm 1.1$ | $72.4 \pm 0.1$ | $36.9 \pm 4.6$ | $68.5 \pm 0.2$ |
| LIMANS$_{4000}$ | **$23.7 \pm 0.5$** | **$69.8 \pm 0.0$** | **$30.8 \pm 0.9$** | **$72.9 \pm 0.3$** | **$35.6 \pm 2.2$** | **$68.2 \pm 0.1$** |

Table 7: Robustness performance of the LIMANS $\ell_2$-attack ($\delta_2 = 0.5$) in terms of RAUD on the CIFAR-10 test data and against the attack detectors plugged in both standard classifier (S.C.) and robust classifier (R.C.). **The smaller the RAUD, the more robust the adversarial attack is**. The best performance is marked in red bold font.

| Detectors d | $d_{PGD}$ | | $d_{Autoattack}$ | | $d_{LIMANS_{10}}$ | |
|---|---|---|---|---|---|---|
| Classifiers $f$ | S.C. | R.C | S.C. | R.C | S.C. | R.C |
| SA | 91.1 | 85.1 | 91.1 | 85.1 | 91.1 | 85.1 |
| PGD | $64.0 \pm 1.2$ | $82.4 \pm 0.0$ | $65.3 \pm 0.6$ | $81.8 \pm 0.1$ | $42.9 \pm 0.6$ | $80.1 \pm 0.0$ |
| Autoattack | $63.1 \pm 1.1$ | $82.19 \pm 0.1$ | $65.8 \pm 0.6$ | **$81.1 \pm 0.2$** | **$42.1 \pm 0.3$** | **$79.4 \pm 0.0$** |
| LIMANS$_{10}$ | $83.5 \pm 0.5$ | $87.4 \pm 0.1$ | $82.9 \pm 0.4$ | $88.1 \pm 0.0$ | $86.9 \pm 0.9$ | $87.6 \pm 0.1$ |
| LIMANS$_{500}$ | $62.9 \pm 0.2$ | $84.0 \pm 0.4$ | $64.7 \pm 1.0$ | $83.8 \pm 0.3$ | $51.7 \pm 1.3$ | $81.7 \pm 0.1$ |
| LIMANS$_{1000}$ | $63.7 \pm 0.6$ | $82.7 \pm 0.3$ | $63.6 \pm 0.9$ | $82.6 \pm 0.2$ | $46.8 \pm 0.7$ | $80.1 \pm 0.1$ |
| LIMANS$_{4000}$ | **$62.6 \pm 1.2$** | **$82.16 \pm 0.2$** | **$62.2 \pm 0.6$** | $82.6 \pm 0.3$ | $46.9 \pm 0.2$ | $80.0 \pm 0.2$ |

**Optimizer.** All the Simple-LIMANS experiments' optimization were performed with the Adam optimizer. We found the Adam optimizer to be the best among several ones, as shown in Figure 7. Indeed similar performances can be reached with a different optimizer such as the RMSProp optimizer, but overall the Adam optimizer or one of its variant seems to be a relevant optimization choice.

**Batch-size.** As Figure 7 shows, when using Simple-LIMANS, the batch size $B$ is not a sensitive hyper-parameter. We found that different batch-sizes could end up to similar performances, the only difference reside in the time consumption, higher batch size yield faster computations. During our experiments, the batch size $B$ was set to $B = 256$ during training and $B = 64$ during inference.

# D  ADDITIONAL RESULTS

## D.1  TRAINING AND TEST FOOLING RATES ON CIFAR-10 ON BOTH $\ell_2$ AND $\ell_\infty$ NORM

We provide here the performance of the LIMANS attacks with increasing the number of atoms $M$ from 1 to 4000. Noting that both the performance on training data, as shown in Figure 9, and on test data, as shown in Figure 8 show that LIMANS attacks can bridge the gap of universal attacks and specific attacks.

## D.2  RAUD TABLES WITH STANDARD DEVIATION FOR BOTH $\ell_2$ AND $\ell_\infty$ NORM

Table 6 and table 7 present the RAUD of Simple-LIMANS and the specific baselines for the $\ell_\infty$ and $\ell_2$ norm with different adversarial example detectors d. Along the RAUD is presented its standard deviation over 5 different random seeds. Performances are shown for both the standard and robust classifier on the CIFAR-10 dataset.

### D.3 ADDITIONAL RESULTS ON TRANSFERABILITY

In this section, we report complementary results about the transferability of the adversarial noise space. As is stated in our paper, the learned space under LIMANS $\ell_\infty$-attack possesses powerful transferability across different classifiers, which is further confirmed by results in Table 9 (part 1) and Table 8 (part 2) and Table 11. Besides, the adversarial noise space obtained using the LIMANS $\ell_2$-attack gains also the transferable property as shown in Table 10.

Table 8: Transferability performance of the LIMANS $\ell_\infty$-attacks on Cifar10 ($\epsilon = 8/255$), in terms of fooling rates (FR). The best transferable results are marked in red bold font and the best specific attacking results are marked in black bold font: Part 2.

|  |  | MobileNet | Inception | ResNet50 | DenseNet | VGG | R-r18 | R-wrn-34-10 |
|---|---|---|---|---|---|---|---|---|
|  | AutoAttack | 17.6 | 17.9 | 18.0 | 18.5 | 19.1 | **27.7** | **39.4** |
|  | UAP | 12.4 | 9.9 | 6.6 | 5.3 | 4.4 | 1.7 | 1.3 |
|  | UAPPGD | 27.0 | 21.5 | 12.9 | 11.7 | 13.1 | 2.5 | 2.6 |
|  | TI-FGSM | 7.9 | 6.6 | 6.8 | 7.7 | 8.3 | 17.2 | 21.2 |
|  | VMI-FGSM | 26.8 | 25.2 | 26.1 | 24.3 | 26 | 26.3 | 32.3 |
| R-wrn-34-10 | VNI-FGSM | 30.0 | 27.7 | 28.6 | 26.3 | 27.4 | 26.7 | 32.1 |
|  | NAA | 13.7 | 11.3 | 10.1 | 10.5 | 11.4 | 9.7 | 15.5 |
|  | RAP | 11.5 | 9.4 | 7.7 | 7.7 | 8.4 | 1.5 | 19.5 |
|  | Ours | **84.9** | **76.6** | **72.8** | **68.9** | **64.0** | 23.2 | 21.6 |

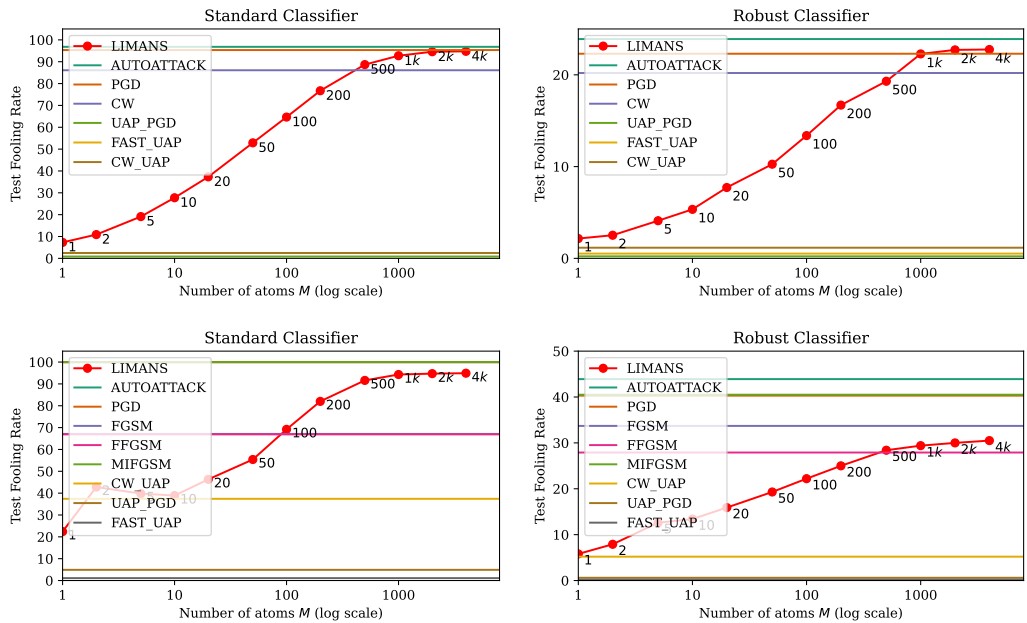

Figure 8: Evolution of LIMANS' test fooling rate according to the number of atoms M. Both specific and universal adversarial attack baselines are shown. The problem is solved under the $\ell_2$ norm constraint (first line) and $\ell_\infty$ norm constraint (second line) on the standard classifier (left figure) and the robust classifier (right figure) on CIFAR-10 using Simple-LIMANS. On average over 5 random seeds the fooling rates vary around 0.4% of FR for the standard model and around 0.1% of FR for the robust model, errorbars are plotted but so tiny, are invisible.

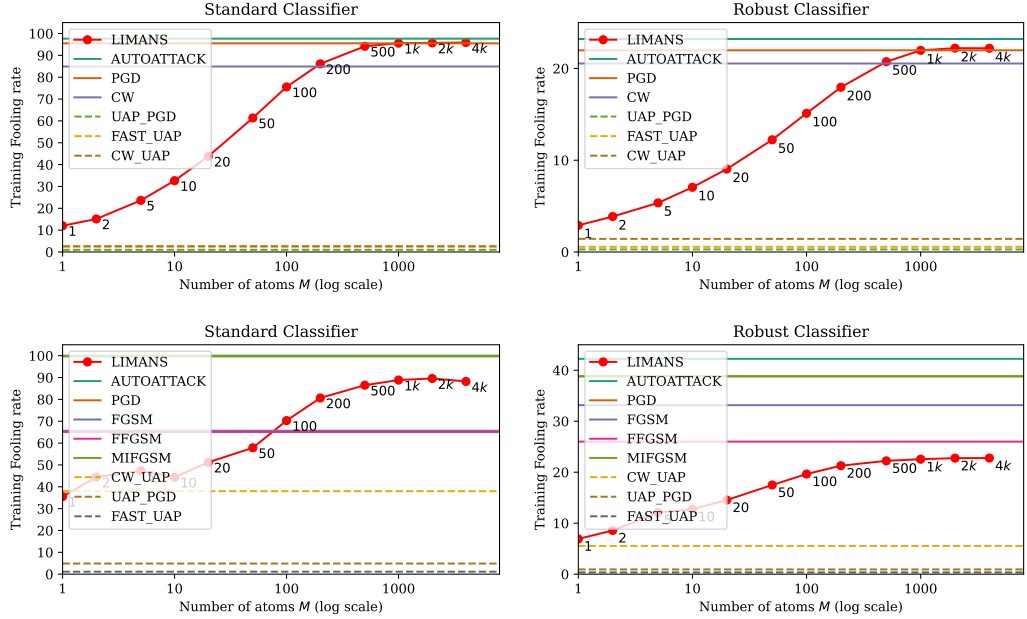

Figure 9: Evolution of LIMANS' training fooling rate according to the number of atoms M. Both specific and universal adversarial attack baselines are shown. The problem is solved under the $\ell_2$ norm constraint (first line) and $\ell_\infty$ norm constraint (second line) on the standard classifier (left figure) and the robust classifier (right figure) on CIFAR-10 using Simple-LIMANS. On average over 5 random seeds the fooling rates vary around 0.4% of FR for the standard model and around 0.1% of FR for the robust model, error bars are plotted but so tiny, are invisible.

Table 9: Transferability performance of the LIMANS $\ell_\infty$-attacks on Cifar10 ($\epsilon = 8/255$), in terms of fooling rates (FR). The best transferable results are marked in red bold font and the best specific attacking results are marked in black bold font: Part 1.

|  |  | MobileNet | Inception | ResNet50 | DenseNet | VGG | R-r18 | R-wrn-34-10 |
|---|---|---|---|---|---|---|---|---|
| MobileNet | AutoAttack | **100** | 87.1 | 37.2 | 32.8 | 22.4 | 1.7 | 1.5 |
|  | UAP | 47.3 | 36.1 | 9.3 | 8.3 | 8.7 | 1.3 | 0.8 |
|  | UAPPGD | 86.2 | 56.1 | 20.5 | 19 | 21.3 | 1.6 | 1.5 |
|  | TI-FGSM | 87.2 | 25.2 | 25.9 | 29.1 | 16.1 | 2.1 | 2.0 |
|  | VMI-FGSM | **100** | 87.3 | 53.1 | 49.8 | 38.9 | 2.2 | 2.5 |
|  | VNI-FGSM | **100** | 88.1 | 54.8 | 53.1 | 40.7 | 2.4 | 2.9 |
|  | NAA | 72.2 | 25.3 | 6.8 | 5.9 | 6.4 | 1.3 | 1.0 |
|  | RAP | 86.7 | 60.3 | 38.5 | 35.7 | 25.8 | 3.0 | 1.6 |
|  | Ours | 97.3 | **92.2** | **73.6** | **66.4** | **67.7** | **10.2** | **10.7** |
| Inception | AutoAttack | 54.7 | **100** | 14.7 | 12.9 | 12.0 | 1.2 | 1.1 |
|  | UAP | 39.2 | 32.9 | 9.3 | 9.7 | 9.4 | 1.5 | 1.1 |
|  | UAPPGD | 73.9 | 75.5 | 26.3 | 23.8 | 27.3 | 2.2 | 1.5 |
|  | TI-FGSM | 19.7 | 60.2 | 19.8 | 21 | 11.4 | 2.2 | 1.5 |
|  | VMI-FGSM | 69.8 | 86.1 | 40.8 | 38.3 | 31.3 | 2.6 | 1.6 |
|  | VNI-FGSM | 75.5 | 89.5 | 44.4 | 42.4 | 36.3 | 3.2 | 2.3 |
|  | NAA | 38.7 | 70.5 | 8.4 | 8.1 | 9.2 | 1.1 | 1.5 |
|  | RAP | 61.9 | 90.2 | 42.0 | 41.7 | 30.3 | 2.3 | 2.7 |
|  | Ours | **98** | 95.1 | **79.6** | **73.9** | **75.8** | **10.7** | **10.8** |
| ResNet50 | AutoAttack | 63.3 | 52.6 | **100** | 54.6 | 25.1 | 1.2 | 2.4 |
|  | UAP | 31.4 | 23.6 | 12.1 | 12.5 | 11.2 | 1.3 | 1.7 |
|  | UAPPGD | 63.3 | 49.4 | 39.4 | 35.1 | 26.1 | 1.1 | 2.3 |
|  | TI-FGSM | 18.4 | 17.1 | 74.0 | 38.5 | 20.4 | 2.2 | 3.0 |
|  | VMI-FGSM | 74.9 | 75.3 | 96.0 | 78.1 | 53.5 | 2.1 | 3.2 |
|  | VNI-FGSM | 78.3 | 76.9 | 95.9 | 80.3 | 57.2 | 2.7 | 2.1 |
|  | NAA | 50.7 | 38.6 | 64.7 | 22.9 | 18.4 | 1.4 | 2.1 |
|  | RAP | 49.0 | 45.7 | 75.1 | 52.5 | 35.4 | 1.6 | 2.8 |
|  | Ours | **96.0** | **92.9** | 91.3 | **81.8** | **82.1** | **11.7** | **13.2** |
| DenseNet | AutoAttack | 56.9 | 51.6 | 48.8 | **100** | 21.8 | 2.1 | 2.0 |
|  | UAP | 27.6 | 20.6 | 10.6 | 12.8 | 11.4 | 1.6 | 1.4 |
|  | UAPPGD | 61.1 | 49.9 | 29.3 | 47.4 | 27.3 | 2.7 | 2.1 |
|  | TI-FGSM | 17.4 | 15.8 | 26.3 | 65.2 | 17 | 2.9 | 2.3 |
|  | VMI-FGSM | 73.7 | 71.8 | 77.2 | 93.1 | 47.9 | 3.3 | 3.7 |
|  | VNI-FGSM | 78.1 | 76.2 | 79.5 | 94.0 | 53.3 | 3.5 | 4.2 |
|  | NAA | 37.2 | 31.1 | 23.7 | 74.9 | 12.5 | 1.2 | 1.5 |
|  | RAP | 47.8 | 43.5 | 48.7 | 75.9 | 35.6 | 3.2 | 3.5 |
|  | Ours | **96.7** | **93.5** | **88.4** | 85.5 | **82.7** | **12.3** | **13.4** |
| VGG | AutoAttack | 62.5 | 58.0 | 43.0 | 44.0 | **100** | 2.7 | 2.7 |
|  | UAP | 22.0 | 18.4 | 10.2 | 10.2 | 10.0 | 1.1 | 1.3 |
|  | UAPPGD | 63.6 | 55.9 | 27.6 | 29.4 | 41.9 | 3.1 | 2.1 |
|  | TI-FGSM | 19.7 | 16.7 | 25.6 | 27.6 | 74.4 | 3.7 | 2.2 |
|  | VMI-FGSM | 66.2 | 64.2 | 57.5 | 56.9 | 96.5 | 3.0 | 2.6 |
|  | VNI-FGSM | 69.3 | 68 | 62.6 | 61.4 | 96.5 | 3.0 | 2.6 |
|  | NAA | 42.3 | 38.3 | 14.5 | 1.8 | 71.6 | 1.6 | 1.2 |
|  | RAP | 46.5 | 44.5 | 39.5 | 40.9 | 73.8. | 3.3 | 3.4 |
|  | Ours | **97.4** | **95.1** | **87.5** | **81.5** | 91.0 | **11.5** | **12.6** |
| R-r18 | AutoAttack | 17.5 | 15.7 | 17.2 | 15.6 | 17.5 | **44.3** | **23.4** |
|  | UAP | 14.5 | 9.5 | 7.1 | 6.4 | 7.6 | 1.9 | 2.6 |
|  | UAPPGD | 18.6 | 13.3 | 9.7 | 8.6 | 10.5 | 3.1 | 3.5 |
|  | TI-FGSM | 8.4 | 5.5 | 8.2 | 7.8 | 8.6 | 26.2 | 13.1 |
|  | VMI-FGSM | 24 | 22.9 | 24.2 | 21.9 | 24.8 | 38 | 22.7 |
|  | VNI-FGSM | 27.1 | 23.1 | 25.4 | 23.8 | 25.6 | 38.1 | 22.9 |
|  | NAA | 16.2 | 11.5 | 11.2 | 10.4 | 10.4 | 18.7 | 7.2 |
|  | RAP | 10.9 | 8.4 | 7.9 | 8.9 | 9.7 | 23.8 | 12.2 |
|  | Ours | **81.3** | **73.2** | **71.7** | **68.3** | **61.7** | 25.3 | 21.6 |

Table 10: Transferability performance of the LIMANS $\ell_2$-attacks on Cifar10 ($\epsilon = 0.5$), in terms of fooling rates (FR). The best transferable results are marked in red bold font and the best specific attacking results are marked in black bold font.

|  |  | MobileNet | Inception | ResNet50 | DenseNet | VGG | R-r18 | R-wrn-34-10 |
|---|---|---|---|---|---|---|---|---|
| MobileNet | AutoAttack | **100** | 50.20 | 14.20 | 13.30 | 8.20 | 0.90 | 0.50 |
|  | UAP | 7.50 | 5.20 | 3.00 | 2.50 | 2.40 | 0.30 | 0.40 |
|  | UAPPGD | 37.90 | 15.20 | 2.00 | 1.10 | 0.90 | 0.30 | 0.20 |
|  | CW | 97.50 | 11.00 | 4.20 | 3.20 | 2.40 | 0.30 | 0.00 |
|  | RAP | 67.30 | 11.20 | 4.20 | 3.90 | 2.60 | 0.50 | 0.10 |
|  | Ours | 95.40 | **91.50** | **61.70** | **59.30** | **51.50** | **4.60** | **5.00** |
| Inception | AutoAttack | 32.80 | **100** | 6.60 | 7.90 | 5.50 | 0.50 | 0.50 |
|  | UAP | 9.80 | 7.50 | 2.50 | 3.50 | 2.90 | 0.20 | 0.10 |
|  | UAPPGD | 26.90 | 16.70 | 1.30 | 2.30 | 2.00 | 0.30 | 0.10 |
|  | CW | 16.30 | 82.80 | 5.00 | 5.20 | 3.70 | 0.30 | 0.00 |
|  | RAP | 13.60 | 43.50 | 3.50 | 3.60 | 2.70 | 0.40 | 0.30 |
|  | Ours | **94.60** | 94.10 | **64.30** | **63.90** | **57.20** | **5.10** | **5.20** |
| ResNet50 | AutoAttack | 31.00 | 23.40 | **99.70** | 26.10 | 10.00 | 1.20 | 0.70 |
|  | UAP | 5.10 | 3.80 | 2.40 | 1.90 | 2.80 | 0.50 | 0.30 |
|  | UAPPGD | 4.10 | 3.20 | 2.20 | 2.30 | 2.20 | 0.40 | 0.20 |
|  | CW | 13.50 | 9.80 | 82.40 | 13.10 | 6.10 | 0.50 | 0.40 |
|  | RAP | 10.20 | 8.60 | 33.00 | 8.60 | 4.90 | 0.40 | 0.30 |
|  | Ours | **92.60** | **87.50** | 78.10 | **71.70** | **61.70** | **7.90** | **7.50** |
| DenseNet | AutoAttack | 32.60 | 25.20 | 27.30 | **99.50** | 10.20 | 0.50 | 0.50 |
|  | UAP | 4.90 | 3.70 | 2.60 | 3.30 | 1.90 | 0.20 | 0.20 |
|  | UAPPGD | 5.00 | 4.50 | 3.20 | 3.70 | 2.10 | 0.20 | 0.20 |
|  | CW | 14.60 | 13.70 | 14.80 | 80.00 | 6.40 | 0.40 | 0.30 |
|  | RAP | 8.00 | 7.30 | 7.70 | 29.00 | 4.50 | 0.30 | 0.30 |
|  | Ours | **91.10** | **87.60** | **74.00** | 74.10 | **62.70** | **8.40** | **7.70** |
| VGG | AutoAttack | 32.00 | 28.20 | 19.50 | 21.10 | **98.90** | 0.80 | 0.60 |
|  | UAP | 4.70 | 3.80 | 2.20 | 2.70 | 2.00 | 0.50 | 0.40 |
|  | UAPPGD | 4.80 | 5.60 | 2.00 | 2.70 | 2.80 | 0.40 | 0.40 |
|  | CW | 10.00 | 8.20 | 5.70 | 7.40 | 79.10 | 0.60 | 0.30 |
|  | RAP | 8.80 | 7.10 | 5.20 | 6.50 | 32.10 | 0.30 | 0.50 |
|  | Ours | **94.20** | **89.00** | **74.80** | **71.00** | 71.70 | **8.00** | **7.10** |
| R-r18 | AutoAttack | 6.70 | 8.30 | 8.00 | 8.50 | 9.60 | **24.60** | 11.00 |
|  | UAP | 3.40 | 3.10 | 2.50 | 2.30 | 1.80 | 0.50 | 0.40 |
|  | UAPPGD | 2.70 | 2.10 | 2.00 | 1.70 | 2.60 | 0.30 | 0.10 |
|  | CW | 9.50 | 11.60 | 10.60 | 10.00 | 11.60 | 22.90 | 3.90 |
|  | RAP | 8.70 | 7.70 | 7.60 | 8.10 | 9.60 | 10.70 | 4.50 |
|  | Ours | **58.70** | **53.80** | **50.30** | **50.70** | **41.80** | 17.60 | **14.60** |
| R-wrn-34-10 | AutoAttack | 7.70 | 7.80 | 8.20 | 7.80 | 8.90 | 15.20 | **22.50** |
|  | UAP | 3.00 | 3.10 | 2.40 | 2.90 | 2.60 | 0.90 | 0.40 |
|  | UAPPGD | 2.90 | 2.80 | 2.20 | 1.00 | 1.60 | 0.70 | 0.60 |
|  | CW | 10.30 | 9.10 | 13.00 | 10.60 | 10.40 | 8.80 | 21.20 |
|  | RAP | 8.10 | 7.30 | 8.10 | 7.60 | 7.70 | 7.10 | 9.90 |
|  | Ours | **59.10** | **54.80** | **51.80** | **50.00** | **42.50** | **17.00** | 14.70 |

Table 11: Transferability performance of the LIMANS $\ell_\infty$-attacks on ImageNet ($\epsilon = 4/255$), in terms of fooling rates. The best transferable results are marked in red bold font, and the best specific attacking results are marked in black bold font.

|  |  | MobileNet | ResNet18 | DenseNet | VGG | R-r18 | R-50-2 |
|---|---|---|---|---|---|---|---|
| MobileNet | AutoAttack | **100** | 26.38 | 20.44 | 26.94 | 1.64 | 1.24 |
|  | UAP | 48.48 | 11.5 | 10.46 | 17.28 | 1.8 | 0.84 |
|  | UAPPGD | 69.94 | 18.04 | 14.34 | 22.34 | 2.72 | 1.56 |
|  | TI-FGSM | 99.74 | 36.98 | 31.66 | 31.24 | 3.2 | 2.56 |
|  | VMI-FGSM | **100** | 44.84 | 37.92 | 42.92 | 2.92 | 2.04 |
|  | VNI-FGSM | 99.98 | 44.64 | 36.54 | 43.62 | 2.88 | 2.00 |
|  | NAA | 84.56 | 15.1 | 11.72 | 16.88 | 2.1 | 1.2 |
|  | RAP | 96.52 | _**54.58**_ | _**47.24**_ | _**49.16**_ | 3.72 | 3.16 |
|  | Ours | 75.24 | 50.06 | 46.94 | 44.34 | _**10.02**_ | _**5.62**_ |
| ResNet18 | AutoAttack | 40.3 | **100** | 35.76 | 34.9 | 1.8 | 1.34 |
|  | UAP | 13.34 | 11.3 | 9.00 | 11.72 | 1.36 | 0.86 |
|  | UAPPGD | 25.3 | 47.22 | 18.44 | 23.26 | 2.5 | 1.44 |
|  | TI-FGSM | 32.06 | 99.84 | 31.38 | 31.66 | 2.98 | 2.8 |
|  | VMI-FGSM | 56.5 | **100** | 51.78 | 50.2 | 2.9 | 2.04 |
|  | VNI-FGSM | 56.74 | 99.98 | 51.4 | _**51.42**_ | 2.84 | 2.04 |
|  | NAA | 22.54 | 97.94 | 14.84 | 19.3 | 2.12 | 1.2 |
|  | RAP | 53.36 | 96.74 | 51.30 | 50.60 | 3.80 | 3.14 |
|  | Ours | _**59.16**_ | 59.16 | _**53.14**_ | 48.28 | _**10.48**_ | _**6.62**_ |
| DenseNet | AutoAttack | 37.72 | 40.4 | **100** | 30.22 | 1.8 | 1.3 |
|  | UAP | 12.76 | 9.94 | 9.8 | 11.42 | 1.24 | 0.92 |
|  | UAPPGD | 22.72 | 20.7 | 40.04 | 20.18 | 2.48 | 1.28 |
|  | TI-FGSM | 30.1 | 35.56 | 99.66 | 27 | 3.12 | 2.32 |
|  | VMI-FGSM | 52.22 | 55.44 | 99.98 | 44.82 | 2.9 | 2.06 |
|  | VNI-FGSM | 53.88 | _**56.9**_ | 99.98 | 46.16 | 2.64 | 2.1 |
|  | NAA | 24.22 | 25.68 | 98.34 | 21.38 | 1.34 | 1.42 |
|  | RAP | 48.16 | 54.12 | 96.76 | 42.00 | 3.12 | 3.30 |
|  | Ours | _**58.86**_ | _**56.9**_ | 57.26 | _**47.74**_ | _**11.3**_ | _**7.32**_ |
| VGG | AutoAttack | 47.94 | 40.06 | 32.62 | **100** | 2.34 | 1.42 |
|  | UAP | 13.34 | 9.8 | 8.82 | 13.6 | 1.34 | 0.78 |
|  | UAPPGD | 24.42 | 23.16 | 18.12 | 46.26 | 2.54 | 1.6 |
|  | TI-FGSM | 33.2 | 38.26 | 29.3 | 99.4 | 2.96 | 2.28 |
|  | VMI-FGSM | 57.52 | 53.46 | 43.76 | 99.86 | 2.9 | 2.2 |
|  | VNI-FGSM | _**57.98**_ | 53.96 | 42.88 | 99.84 | 2.76 | 2.24 |
|  | NAA | 19.62 | 14.92 | 12.18 | 79.96 | 2.18 | 1.4 |
|  | RAP | 53.14 | 53.12 | 42.68 | 95.68 | 3.48 | 2.84 |
|  | Ours | 57.68 | _**54.14**_ | _**50.04**_ | 51.62 | _**10.68**_ | _**6.24**_ |
| R-r18 | AutoAttack | 13.7 | 15.8 | 10.82 | 14.6 | **71.74** | 10.78 |
|  | UAP | 11.52 | 9.32 | 8.46 | 10.90 | 1.44 | 1.16 |
|  | UAPPGD | 14.00 | 12.34 | 11.20 | 13.56 | 3.14 | 1.66 |
|  | TI-FGSM | 11.88 | 13.42 | 10.08 | 11.02 | 54.46 | 10.14 |
|  | VMI-FGSM | 17.00 | 17.80 | 12.12 | 16.08 | 64.98 | 11.94 |
|  | VNI-FGSM | 16.14 | 17.66 | 12.48 | 16.08 | 63.22 | 11.74 |
|  | NAA | 11.46 | 10.86 | 9.34 | 11.42 | 21.48 | 4.9 |
|  | RAP | 11.32 | 10.80 | 8.16 | 10.32 | 45.80 | 7.94 |
|  | Ours | _**37.14**_ | _**33.2**_ | _**33.76**_ | _**29.90**_ | 29.84 | _**12.94**_ |
| R-50-2 | AutoAttack | 20.14 | 22.76 | 17.36 | 19.44 | 15.42 | **59.02** |
|  | UAP | 9.88 | 7.60 | 6.96 | 8.62 | 1.84 | 1.24 |
|  | UAPPGD | 14.54 | 12.56 | 10.92 | 14.36 | 2.16 | 1.38 |
|  | TI-FGSM | 14.16 | 16.34 | 12.68 | 13.68 | 17.16 | 43.66 |
|  | VMI-FGSM | 24.22 | 26.66 | 20.12 | 23.86 | 17.82 | 54.56 |
|  | VNI-FGSM | 23.88 | 26.22 | 19.68 | 23.28 | 18.00 | 52.28 |
|  | NAA | 14.08 | 13.12 | 10.20 | 14.04 | 9.82 | 12.58 |
|  | RAP | 13.82 | 14.06 | 10.52 | 13.5 | 15.54 | 34.1 |
|  | Ours | _**42.18**_ | _**42.5**_ | _**42.46**_ | _**34.22**_ | _**23.7**_ | 18.02 |