# OpenReview forum: "LIMANS: Linear Model of the Adversarial Noise Space"
_ICLR.cc/2024/Conference — Submitted to ICLR 2024_

### Official Review · Reviewer_jQoc · 2023-10-31

**Soundness:** 2 fair
**Presentation:** 3 good
**Contribution:** 3 good
**Rating:** 6
**Confidence:** 4

**Summary:**

The paper introduces LIMANS, a new way to model adversarial perturbations as linear combinations of universal adversarial directions. A white-box adversarial attack is developed based of the proposed formalism. The attack and proposed modeling aim to bridge the gap between universal perturbations (high transferability, low attack success) and instance-based attacks (high attack success, low transferability). Two stochastic gradient-based algorithms for learning universal adversarial directions are proposed. Experiments are performed on CIFAR-10 and ImageNet, comparing LIMANS against other white-box attacks.

**Strengths:**

- The idea of unifying universal and custom adversarial examples seems novel and interesting.
- LIMANS exhibits improved attack transferability and success rate when detectors are used.

**Weaknesses:**

# Lack of in-depth analysis

- The proposed method, optimization objectives, relaxations, etc., are introduced as they are, without any small or large scale analyses or too many explanations.
- The paper does not show the universality of the adversarial directions computed by LIMANS, other than through attack transferability. Each input point seems to get its own atom in the learned parameters, so there is no reason to believe that they are universal.
- When studying transferability of adversarial examples, one should arguably look beyond universal perturbations and also consider black-box attacks. These are some of the attacks that offer improved transferability by default. While black-box attacks are listed as an item for future work, to me they are well within the scope of this paper. [Boundary attack](https://openreview.net/pdf?id=SyZI0GWCZ) in particular seems relevant to the motivation presented around stability of decision boundaries per dataset.

# Relation to prior work

- The adversarial ML community has seen many ideas and frameworks for explaining adversarial examples. Many of these were disproven since, and the ones that still hold provide only partial explanations. The paper cites very few (if any) of these results and does not set the current work in the appropriate context. Coupled with the prior point on the depth of the proposed analysis, I am not convinced by the potential significance and impact of the proposed results.

# Potential performance limitations

- A significant practical limitation of the proposed method is that it needs access to at least a thousand clean samples in order to achieve performance remotely close to state-of-the-art attacks in white-box.
- Another limitation that is not addressed in the paper is the cost of crafting adversarial examples with LIMANS after fitting the initial parameters. It seems parameters need to be fitted for each new data point.

# Clarity

- The presentation and typography of the paper could be improved, please see some suggestions among minor points. The paper could also use additional proofreading.

# Minor points

- Consider using `\citep` when the cited references are not part of a sentence for proper citation formatting (see [here](https://www.overleaf.com/learn/latex/Natbib_citation_styles)).
- Repeat sentence fragment in Contributions: "The associated optimization problem, allowing to learn"
- "the decision boundaries of different classifiers are closed" -> use "close" instead of "closed".
- Some elements of the ICLR style are missing from the draft, like the document header. The header sizes also differ. The authors should ensure they use the ICLR style.

**Questions:**

1. It would be interesting to see to which extent the learned directions vary, e.g., when the data sample changes. Are the learned atoms relatively stable or indeed universal?
2. How does the present work relate to the other theories trying to explain the existence of adversarial examples (e.g., based on robust and non robust features in the data [[Ilyas et al., 2019](https://arxiv.org/pdf/1905.02175.pdf)] [[Kim et al., 2021](https://proceedings.neurips.cc/paper/2021/file/8e5e15c4e6d09c8333a17843461041a9-Paper.pdf)], the structure of adversarial spaces [[He et al., 2018](https://arxiv.org/pdf/1812.01198.pdf)], [[Paiton et al., 2022](https://openreview.net/forum?id=2p_5F9sHN9)], [[Sheatsley et al., 2023](https://www.usenix.org/system/files/sec23summer_256-sheatsley-prepub.pdf)])?

---

> ### Author Response · Authors · 2023-11-21
>
> ### Lack of in-depth analysis
> > 1. The proposed method, [...] many explanations.
>
> Since very similar objectives and relaxations are commonly used in the optimization community for different purposes, we have judged not necessary to include an in-depth analysis. We suggest adding one motivating line and citations to guide the reader. Nonetheless, the proposed method has been evaluated on small scale (CIFAR-10) and large scale (Imagenet) datasets for various architectures.
>
> > 2. The paper [...] universal.
>
> The learned directions aim to span the space of the adversarial noise, which is related to the original data space and to some extent, the classifier. As illustrated in the figures above, a two-atom dictionary can construct all the adversarial examples, if they exist. This space is indeed universal. Moreover, through the observed transferability across different classifiers, this conclusion is further confirmed. When attacking a given input, the composition coefficient needs to be computed.
>
> > 3. When studying [...] per dataset.
>
> In black-box attacks, transfer-based methods play a crucial role. To attain strong performance in black-box scenarios, preliminary research on the transferability of a model is vital. This constitutes the primary objective of our current work. We analyze the properties of the learned space to subsequently apply it to enhance black-box attack strategies.
>
> ### Relation to prior work
> > 1. The adversarial [...] proposed results.
>
> Please refer to the response to weakness 1 of reviewer XSE8.
>
> ### Potential performance limitations
> > 1.A significant practical [...] attacks in white-box.
>
> You are right, in order to optimize LIMANS' dictionary, we need at least few clean samples and a source classifier to convergence towards optimized fooling parameters. However one could also argue that most of the black-box attack do impose the same constraint, also white-box attacks have their hyper-parameters that also need to be tuned. Therefore we believe the need of a training set not to be too limiting compared to other propositions.
>
>
> > 2. Another [...] data point.
>
> Thank you for your valuable question. The goal of our proposed method is not only to construct adversarial examples but also to uncover common mechanisms within a dataset for crafting adversarial noises.
>
> ### Clarity
> >1. The presentation [...] proofreading.
>
> Thank you for your valuable insights and guidance in enhancing our paper.
>
> ### Minor points
> > 1. Consider using \citep [...] formatting (see here).
>
> Thank you for the advice; it has been incorporated into the modifications.
>
> > 2. Repeat sentence [...] of "closed".
>
> Thank you, we have corrected it.
>
> > 3. Some elements of the ICLR [...] style.
>
> Thank you, we have aligned it precisely with the ICLR style.
>
> ### Questions
>
> > 1. It would be interesting [...] universal?
>
> It is indeed an intriguing direction. Currently, we are exploring the universality of adversarial directions across the entire dataset rather than focusing on a single example. We plan to delve deeper into the universality of a subspace corresponding to an input.
>
> > 2. How does the present work [...] [Sheatsley et al., 2023])?
>
> Similar to other research efforts, LIMANS endeavors to uncover the intrinsic mechanisms behind adversarial attacks and ultimately enhance the robustness of the classifier.
> While our work doesn't explicitly manipulate the feature space, it is supposed to fortify non-robust features discussed in [Ilyas et al., 2019] [Kim et al., 2021].
> Prior studies on the structure of the adversarial space have predominantly focused on analyzing the local adversarial space around an input [He et al., 2018], [Paiton et al., 2022], [Sheatsley et al., 2023]. In contrast, our space encompasses the entire dataset.Moreover, through global optimization on the entire dataset, the learned adversarial directions exhibit less noise-dependent and architecture-dependent characteristics (as defined in [He et al., 2018]), but instead, they possess a more data-dependent composition. This introduces the concept of transferability in the learned space. We also affirm the conclusion in [Paiton et al., 2022] that robust classifiers are situated in higher dimensions.

---

### Official Review · Reviewer_APAo · 2023-10-31

**Soundness:** 2 fair
**Presentation:** 2 fair
**Contribution:** 3 good
**Rating:** 5
**Confidence:** 4

**Summary:**

This work proposes LIMANS, a process for creating adversarial examples via a learned linear combination of adversarial atoms (noise patterns). The motivation for this work is to explicitly define a process to vary the specificity of an adversarial perturbation to a particular input vs the ability of the perturbation to be successfully applied to multiple inputs. The efficacy of this work is demonstrated by producing adversarial examples (composed of differently weighted 'atoms' learned via one of two provided algorithms) on several different model architectures trained on CIFAR-10 and Imagenet. The ability of these same atoms to be used on other architectures (while still learning the correct weights of these atoms) is evaluated, along with performance against an adversarial example detector. As a result, this work is able to explore the dimensionality of adversarial spaces in CIFAR-10 and Imagenet trained DNN classifiers, and how it is shared across different architectures trained on the same dataset.

**Strengths:**

* (Major) This work shows each contribution of LIMANS stated in the introduction via sound experiments on CIFAR-10 and Imagenet, with multiple DNN architectures, and against a detector. With some exceptions detailed in questions, the results seem convincing.

* (Moderate) Figure 2 does a good job in how varying example-specificity affects the attack success rate.

* (Moderate) The motivation for this work is interesting, and could lead to better understanding of different causes of DNN vulnerability.

**Weaknesses:**

* (Major) The main results show the superiority of LIMANS transferability over example-specific attacks in Table 2 and Table 3. However, this does not quite seem like a fair comparison since, in my understanding, LIMANS is allowed to adjust $v$ for each example on the other architectures whereas the other example-specific attacks are not allowed to adapt at all. Therefore, it is not really surprising that an attack allowed to adapt in any way would have a much higher attack success rate.

* (Moderate) At least in the main text, this work omits a full description of how the learned "atoms" are calculated. Specifically, the text mentions that the atoms may not be suitable (i.e., $D$ not in $\mathcal{D}$) after Algorithm 2, but only says some post-processing is done to fix it. A description of this post-processing and discussion on how it retains the desired qualities of $D$ is necessary.

* (Moderate) This work has several typos or critical missing words that introduce ambiguity on what is meant and makes this paper more difficult to read (shapes of variables are inconsistent, some variables are undefined, incomplete sentences, missing critical words e.g., writing "1" instead of "Algorithm 1" or "Figure 1", missing periods in between sentences, missing letters (FUTUR -> FUTURE), no space before some sentences). More details in questions.

* (Moderate) The baselines for universal attacks seem low when comparing to values in prior work. For example, Figure 2 shows three different versions of UAP getting near 0% success rate, but the original works get near 90% success rate on the same dataset (CIFAR-10) and similar architectures (VGG11 vs VGG16).

**Questions:**

* Am I correct in my understanding that LIMANS is allowed to adjust $v$ via gradient feedback from unseen model architectures to enhance the "transferability" measurements in Tables 2 and 3? If so, is it the case that the other baselines (e.g., VNI-FGSM, NAA, RAP) were not allowed to have this same feedback?

* In equation 1, the subscript $p$ seems to denote the type of norm, but the upper case P represents the dimensionality of the input. Is this correct? If so, it would be best to change one of these to avoid confusion.

* What is capital V in algorithms 1 and 2? is it the same as lower case v? If so, why is it initialized to a shape of $\mathcal{N}(0, 1_{P \times M})$ when $v$ is earlier said to be a vector of size $M$ for each example?

* Why do the universal attacks have low success rates in this work compared to their higher success rates in the original works given the same dataset and similar architectures?

* Section 3 says that Simple-LIMANS may produce a $D$ that is not in $\mathcal{D}$ (i.e., does not fit the dictionary requirements). The work mentions that some post-processing is used to fix this. What is this post-processing and how does it ensure the atoms are still universal adversarial directions? Also, how does this post-processing ensure that later $v$s calculated on unseen examples can still work with this modified $D$? It could be the case that one atom in the original $D$ had a large magnitude (outside $l_p$ constraints) that allowed a small corresponding element of $v$ to have a large effect. Assuming post-processing reduces this magnitude of this atom of $D$, how can a $v$ calculated on unseen examples be able to have the same effect?

* What is the reasoning for why LIMANS is able to evade detection better in Table 1? I did not notice any discussion on this trend or a justification of why LIMANS would have this advantage.

* In Table 1, what is meant by Detectors $d$ having different attack names as subscripts to them such as $d_{\text{FGSM}}$? Is this detector tuned to defend against FGSM? If so, why is there only one detector for $LIMANS_{10}$ and not one for the claimed strongest attack of $\text{LIMANS}_{4000}$? What is SA (standard accuracy?)?

* Can you provide a citation for this statement in Section 4.2 please? "Moreover, for robust classifiers, the decision boundaries are more complicated..." In my understanding, and as evidenced by the pictures in Figure 3, it seems that the decision boundaries are smoother and less complicated looking.

* Could you provide citations for this sentence near the beginning of section 2: "However, it has been found that they are not as effective in fooling other classifiers, namely, adversarial examples yielded by specific attacks are poorly transferable from one classifier to another."

---

> ### Author Response · Authors · 2023-11-21
>
> ### Weaknesses
> - 1.
>
> Our emphasis is on researching the adversarial noise space, a domain contingent upon the dataset space. Our proposed algorithm facilitates the learning of a dictionary that spans either partially or entirely the adversarial noise space, demonstrating transferability. A comparison with transferable attacks confirms that a specific attack is unable to outperform a globally trained adversarial noise space in terms of transferability.
>
> - 2.
>
> This post-processing involves projecting onto the valid space of D, denoted as $\mathcal{D} ={D \,  ||D_j||_p=1, \forall j \in {1,\dots,M}}$. This projection is introduced in step 18 of Algorithm 2. It's noteworthy that we explicitly enforce the constraint on $Dv$ during training rather than on $D$ and $v$ separately. However, this does not impact the final results due to the post-processing step, where $d_i=d_i/||d_i||_p$ and $v_i = v_i*||d_i||_p$ are applied.
>
> - 3.
>
> We have carefully proofread the article to correct these issues.
>
> - 4.
>
> Please refer to our answer to Reviewer S66D. Additionally, we would like to mention that, for the sake of fairness, we selected the most common adversarial budgets, namely $||\epsilon||_2 < 0.5$ and $||\epsilon|| _\infty < 8$ for attacks on CIFAR-10. Of course, choosing larger threshold would permit to yield better fooling performance at the cost of more detectable attacks. We also would like to point that CIFAR-10 proves to be more challenging to attack than ImageNet, as evidenced by the RobustBench (https://robustbench.github.io/) performance metrics for AutoAttack.
>
> ### Questions
> - 1.
>
> Yes, our algorithm is designed to learn a dictionary that endeavors to span the adversarial noise space. To find an adversarial sample for a given input, computation of the composition coefficients is necessary. The comparison highlights that, as depicted in the figures above, specific attacks often struggle to transfer due to the local gradient direction bias among classifiers. In contrast, a universal attack significantly degrades performance when $\vert\vert \epsilon \vert\vert_p$ is small. Conversely, LIMANS, having learned the space of adversarial noise, exhibits improved performance when transferred to different classifiers.
>
> - 2.
>
> Thank you. It has been fixed.
>
> - 3.
>
> In algorithms 1 and 2, the capital V represents the matrix that is the concatenation of all the training coding vectors, while the lower case v represents only one single training coding vector.
>
> - 4.
>
> Please refer to the response addressing Weakness 4.
>
> - 5.
>
> Please refer to the response addressing Weakness 2.
>
> - 6.
>
> It is assumed that the robustness against the detector is achieved due to the optimization-based schema.
>
> - 7.
>
> The name in the subscript of each detector corresponds to the attack used to produce the training set on which the detector is trained. Put differently, d_FGSM is made to detect attacks crafted by FGSM. First, standard accuracy (SA) is defined as the classification accuracy of a DNN model without adversarial examples or attack detectors. Then, robust accuracy (RA) is defined as the classification accuracy of a DNN model with adversarial examples.
>
> - 8.
>
> Imaging a simplest case, a two-class classification problem with a linear classifier; here, a unique direction enables inputs to find an adversarial example if one exists. Consequently, the dimension of the adversarial noise is 1. In more complex scenarios, such as depicted by classifier 2 in the figures above, it is still possible to construct a one-dimensional space. However, when the classifier, like classifier 1 in the figures above, is even more intricate, the dimension of the space should be 2. Therefore, based on the larger dimension of the space for robust classifiers and their slow slope in Figure 2 of the paper, it implies the complexity of the robust classifiers' boundaries. Figure 3 illustrates the image of one direction, which is related to classifier boundaries rather than the classifier itself. In reality, explicitly illustrating the classifier boundaries of deep neural networks is challenging. [Paiton, Dylan M., et al. "The Geometry of Adversarial Subspaces." (2021).] emphasize this through local analysis.
>
> - 9.
>
> Please refer to "existing adversarial attacks have exhibited great effectiveness but with low transferability..." in [Wang X, He K. Enhancing the transferability of adversarial attacks through variance tuning[C]//Proceedings of the IEEE/CVF Conference on Computer Vision and Pattern Recognition. 2021: 1924-1933]. Furthermore, we presented empirical estimates of the transferability of specific attacks, including transferable specific attacks like VNI-FGSM.

---

### Official Review · Reviewer_XSE8 · 2023-11-01

**Soundness:** 3 good
**Presentation:** 3 good
**Contribution:** 3 good
**Rating:** 5
**Confidence:** 3

**Summary:**

The paper introduces LIMANS, a model that bridges the gap between specific and universal adversarial attacks on deep neural network (DNN) classifiers. LIMANS defines an adversarial noise space, enabling specific attacks to be represented as combinations of universal adversarial directions. This approach enhances the efficiency and transferability of adversarial attacks while also providing insights into DNN vulnerabilities.

**Strengths:**

+ The paper is well-written and presents its concepts in a clear and understandable manner.
+ The formulation that bridges the gap between universal and specific adversarial attacks is an intriguing contribution, offering a promising avenue for improving deep neural network security.

**Weaknesses:**

- The study primarily relies on empirical evaluations to support its claims, which may leave room for a more in-depth theoretical exploration of the connection between universal and specific adversarial attacks.
- The results suggest a trade-off between the transferability of attacks to other models and their effectiveness on the source model, potentially requiring further investigation and trade-off analysis.
- The conflicting results observed on different datasets indicate that the proposed approach might not be universally applicable, and its limitations should be carefully considered in practical applications.

**Questions:**

The paper builds upon the notion that specific attacks exhibit poor transferability. However, specific attacks (as known as adversarial examples) are shown to be transferable in prior literature [refA, B]. Thus, it is crucial for the paper to substantiate this assertion with empirical evidence, shedding light on the extent to which specific attacks struggle when transferred.

The paper also posits that the proximity of decision boundaries among multiple DNN classifiers trained on the same dataset suggests potential transferability in the adversarial noise space, which can be applicable for both specific and universal adversarial examples. However, the distinction between specific and universal adversarial attacks and why universals tend to excel in transferability remains somewhat unclear in the current version.

While the paper's formulation is intriguing, reinforcing it with theoretical contributions that establish the linkage and connection between specific and universal adversarial attacks in the adversarial space would enhance its significance.

Notably, the results in Table 3 appear to indicate a trade-off between the proposed approach's transferability and its effectiveness, with better transferability when the source and target models differ but reduced effectiveness in scenarios where they are the same. These findings conflict with those presented in Table 2. Given the empirical nature of the paper, these conflicting results cast some doubt on the validity of the hypothesis and the efficacy of the proposed method.

[refA] Papernot, Nicolas, Patrick McDaniel, and Ian Goodfellow. "Transferability in machine learning: from phenomena to black-box attacks using adversarial samples." arXiv preprint arXiv:1605.07277 (2016).

[refB] Tramèr, Florian, et al. "The space of transferable adversarial examples." arXiv preprint arXiv:1704.03453 (2017).

---

> ### Author Response · Authors · 2023-11-21
>
> ### Weaknesses
> > 1. The study primarily [...] connection between universal and specific adversarial attacks.
>
> We acknowledge that our proposed modeling, and more broadly, related works on adversarial noise space modeling, reveal new research directions that necessitate theoretical guidance and guarantees. In this paper, we have opted for a more practical and empirical evaluation, intending to delve into its theoretical aspects in future works. Nevertheless, we share the following insights that we recommend including in the final version to further justify our approach.
> The overall space of adversarial noise is equivalent to that of the classifier boundaries. However, due to the complexity of the deep neural network, direct analysis of the classifier seemed out of reach. Consequently, we endeavored to elucidate certain characteristics of the space (e.g., its dimension) through empirical experiments. Despite this, a straightforward analysis can be conducted, as illustrated in the figures above. In contrast, a universal attack seeks an average direction across the entire dataset to deceive as many examples as possible. However, with a single direction, it becomes impractical for inputs in case 2 to discern perturbations. LIMANS (case 2) bridges the gap between these two types of attacks. It dispenses with the need to seek a direction for each input but instead focuses on the space spanned by {$d_0$, $d_1$}.
>
> > 2. The results suggest [...] and trade-off analysis.
>
> In fact, the transferability of the space is proportional to its effectiveness on the source model. Typically, a classifier operates in higher dimensions, necessitating more atoms to construct the space. This makes it easier to transfer the space to a target model situated in a lower-dimensional space. For instance, the dictionary learned on Inception can be readily transferred to MobileNet. Conversely, transferability diminishes. Moreover, if the training sets of two models are not in the same space, transferability is significantly affected.
>
> > 3. The conflicting results observed [...] in practical applications.
>
> We disagree and in fact strongly believe that the results obtained on the two datasets are consistent. Indeed, for complex datasets like ImageNet having $C=1000$ classes, learning the complete adversarial noise space is a challenge. Assuming the simplest case where the classifier is linear and its hyperplanes are independent, $C\times (C-1)/2 = (1000 \times 999)/2  = 499500$ atoms are needed to construct the space. In our experiments, we only learned a 100-dimensional subspace of the adversarial noise space, yet it allowed us to deceive half of the examples.  This highlights a practical limitation of our algorithm, as it requires substantial memory to learn a larger, or even complete, dictionary.
>
> ### Questions
> > 1. The paper builds upon the notion [...] specific attacks struggle when transferred.
>
> [Ref B] demonstrates the existence of a transferable subspace around an input, inspiring transfer-based black-box attacks. Specific attack seeks the smallest perturbation that misleads the prediction (case 1). Typically, this tailored perturbation at the border of the adversarial space associated with a source model (in blue) seldom intersects with the adversarial space of another target model (in orange), limiting its transferability. We assess the transferability of the specific attack, such as AutoAttacks, revealing its limitations in the context of transfer. Our proposed method learned the adversarial space of a dataset not a single input, which makes it possible to craft an attack by a linear combinaison of the atoms under certain condition.
>
> > 2. The paper also posits [...] in the current version.
>
> As explained earlier, a specific attack identifies adversarial noise along the direction of the gradient, which is locally perpendicular to the classification boundaries. However, a universal attack searches for the average direction across the entire dataset. Comparing the averaged direction of two classifiers, their local gradient directions differ more.
>
> > 3. While the paper's formulation is intriguing, [...] its significance.
>
> We agree. The analysis previously mentionned has been added to our paper to provide additionnal insights.
>
>
> > 4. Notably, the results in Table 3 [...] efficacy of the proposed method.
>
> Kindly refer to the response addressing Weakness 3.

---

### Official Review · Reviewer_S66D · 2023-11-03

**Soundness:** 2 fair
**Presentation:** 3 good
**Contribution:** 2 fair
**Rating:** 6
**Confidence:** 3

**Summary:**

The authors propose a method to model the adversarial perturbation in a linear combination space, which builds a dictionary to bridge the link between the universal adversarial attack and specific adversarial attack. Experiments on different datasets illustrate the strong attacking performance wrt existing adversarial example detectors, and the learned adversarial noise space brings superior transferability.

**Strengths:**

+ The paper is well-structured and organized, and easy to follow. The methodology and experimental setup are adequately explained.
+ The proposed LIMANS build the bridge between the universal adversarial perturbation and specific adversarial perturbation.
+ Extensive experiments are conducted to support their proposed method. Empirically and theoretically, compared to the previous universal attack methods, the LIMANS is more efficient on the two datasets.

**Weaknesses:**

- The performance of the proposed LIMANS relies on the choice of the source classifier.
- It is not clear the extra computation overhead for learning the model from the source classifier, it would be better the compare the spending time across different attacks (for Simple-LIMANS and Regularized-LIMANS respectively).
- The results of the transferability performance of UAP look much lower than that in the existing literature, even lower than some specific attacks when attacking a standard-trained model. Why?
- Could you explain more about why you only use the validation set and not the training set?
- To my knowledge, you seem to miss some SOTA methods in the field of universal attack (e.g., [1]).
- The simple-LIMANS seems to remove the tunable hyper-parameter lambda of regularized-LIMASNS, therefore also providing the results of regularized-LIMASNS would be desirable.
- A lot of typos/mistakes, e.g. "Table ??", "FOR FUTUR", "learn The associated"...

[1] Generalizing universal adversarial perturbations for deep neural networks, Machine Learning 112 (5), 1597-1626

**Questions:**

Pls see Section Weaknesses.

---

> ### Author Response · Authors · 2023-11-21
>
> > 1. The performance of the proposed LIMANS relies on the choice of the source classifier.
>
> The reviewer is right. By design, the modeling of the adversarial noise space is optimized to fool a single source classifier. However, the results depicted in Tables 2 and 3 show great transferability performance to other classifiers as well. We believe that these results stem from the fact that the proposed framework, in-between specific and universal attacks, allows to learn adversarial patterns that are less dependent on the source classifier but more representative of the data diversity instead.
>
> > 2. It is not clear the extra computation overhead for learning the model from the source classifier, it would be better the compare the spending time across different attacks (for Simple-LIMANS and Regularized-LIMANS respectively).
>
> We agree with the reviewer that a time comparison of both attacks would be interesting. However, these two attacks play different roles. Simple-LIMANS acts as a decent, quick-to-run, and parameter-free attack, while Regularized-LIMANS might show stronger performances when coupled with the right hyper-parameters, at the burden of finding them.
>
> > 3. The results of the transferability performance of UAP look much lower than that in the existing literature, even lower than some specific attacks when attacking a standard-trained model. Why?
>
> In the original work on UAP, its performance is assessed on ImageNet under the condition that $\vert\vert \epsilon \vert\vert_\infty \leq 10$. In our experiments, we impose a more restrictive constraint that is $\vert\vert \epsilon \vert\vert_\infty \leq 4$. This results in a drop of performance, which is overcome by our proposed method, LIMANS. Nonetheless, our work aligns with the conclusions of the existing literature, demonstrating that universal attacks exhibit great universality at the expense of lower fooling performances on the source model.
>
> > 4. Could you explain more about why you only use the validation set and not the training set?
>
> It is assumed that the training set is used to train the classifier, and in practice, it is usually not publicly accessible and we only have access to the validation data.
>
>
> > 5. To my knowledge, you seem to miss some SOTA methods in the field of universal attack (e.g., [1]).
>
> You are right, due to extensive research in the field producing very fast new SOTA methods, this method has unfortunately been missed. We suggest to add the reference in the final version.
>
> > 6. The simple-LIMANS seems to remove the tunable hyper-parameter lambda of regularized-LIMANS, therefore also providing the results of regularized-LIMASNS would be desirable.
>
> Unfortunately, this is not feasible. A direction linked to an input is parallel to $\delta^{*}=\displaystyle \text{argmin}_{\{f(x+\delta)\neq f(x)\}} ~||\delta||_p$. The ensemble of optimal directions globally encompasses all these directions within a dataset. This cannot be achieved with a fixed threshold.
>
> > 7. A lot of typos/mistakes, e.g. "Table ??", "FOR FUTUR", "learn The associated"...
>
> It has been fixed, thank you for pointing them.

---

### Meta-Review · Area_Chair_wpvS · 2023-12-11

**Metareview:**

The reviewers appreciated the contribution, but the paper is not quite ready for publication. There are serious presentation issues, which the reviewers pointed out as actually hindering assessing the contribution; this is not what one would expect from an ICLR submission.

Beyond this, reviewers remained concerned on several fronts: theoretical guarantees, and a connection to works that indicate that transferability is possible, would strengthen the paper. So would arguing about why the classifier-dependence of LIMANS is not a problem. The "fairness" over competitors in Tables 2 and 3 remains in doubt. The level of scholarship, particularly w.r.t. sota adversarial attacks, could be improved. Table 1 is also hard to interpret: it is not clear why LIMANS is able to evade detection better than competitors in Table 1; it is also not clear why detectors are tuned to an attack and why, if so, there is no detector for LIMANS_4000.

**Justification For Why Not Higher Score:**

Reviewers raised numerous concerns, and were not convinced by the rebuttal. The paper is also unpolished, not at the level expected from an ICLR submission.

**Justification For Why Not Lower Score:**

N/A

---

### Decision · Program_Chairs · 2024-01-16

Reject